# Two neuronal peptides encoded from a single transcript regulate mitochondrial complex III in *Drosophila*

Justin A Bosch[1]*, Berrak Ugur[2†], Israel Pichardo-Casas[1], Jordan Rabasco[1], Felipe Escobedo[1], Zhongyuan Zuo[2], Ben Brown[3], Susan Celniker[3], David A Sinclair[1], Hugo J Bellen[2,4,5,6], Norbert Perrimon[1,6]*

[1]Department of Genetics, Blavatnik Institute, Harvard Medical School, Boston, United States; [2]Department of Molecular and Human Genetics, Baylor College of Medicine, Houston, United States; [3]Lawrence Berkeley National Laboratory, Berkeley, United States; [4]Jan and Dan Duncan Neurological Research Institute, Texas Children's Hospital, Houston, United States; [5]Department of Neuroscience, Baylor College of Medicine, Houston, United States; [6]Howard Hughes Medical Institute, Houston, United States

**\*For correspondence:**
jabosch@hms.harvard.edu (JAB);
perrimon@genetics.med.
harvard.edu (NP)

**Present address:** †Departments of Neuroscience and Cell Biology, Howard Hughes Medical Institute, Yale University School of Medicine, New Haven, United States

**Abstract** Naturally produced peptides (<100 amino acids) are important regulators of physiology, development, and metabolism. Recent studies have predicted that thousands of peptides may be translated from transcripts containing small open-reading frames (smORFs). Here, we describe two peptides in *Drosophila* encoded by conserved smORFs, Sloth1 and Sloth2. These peptides are translated from the same bicistronic transcript and share sequence similarities, suggesting that they encode paralogs. Yet, Sloth1 and Sloth2 are not functionally redundant, and loss of either peptide causes animal lethality, reduced neuronal function, impaired mitochondrial function, and neurodegeneration. We provide evidence that Sloth1/2 are highly expressed in neurons, imported to mitochondria, and regulate mitochondrial complex III assembly. These results suggest that phenotypic analysis of smORF genes in *Drosophila* can provide a wealth of information on the biological functions of this poorly characterized class of genes.

## Editor's evaluation

The paper identifies two small protein products, Sloth1 and Sloth2, that form a complex and perform an important role in respiratory metabolism. These important findings contribute to the functional understanding of small proteins as well as to broadening the knowledge of the physiology of mitochondria, especially in the tissues of high energy demand and unique metabolic profiles such as neurons. Convincing and rich evidence is provided that Sloth1/2 is involved in maintaining the functionality of the respiratory chain, specifically of Complex III.

## Introduction

Naturally produced peptides are regulators of metabolism, development, and physiology. Well-known examples include secreted peptides that act as hormones (*Pearson et al., 1993*), signaling ligands (*Katsir et al., 2011*), or neurotransmitters (*Snyder and Innis, 1979*). This set of peptides are produced by cleavage of larger precursor proteins (*Fricker, 2005*), but peptides can also be directly translated from a transcript with a small open-reading frame (smORF) (*Couso and Patraquim, 2017*; *Plaza et al., 2017*; *Hsu and Benfey, 2018*; *Yeasmin et al., 2018*). Due to their small size (<100 codons), smORFs

have been understudied. For example, smORFs are underrepresented in genome annotations (*Basrai et al., 1997*), are theoretically a poor target for EMS mutagenesis, and are often ignored in proteomic screens. Consequently, there is growing interest in this class of protein-coding gene as a potentially rich source of novel bioactive peptides (*Mudge et al., 2022*).

A major obstacle in identifying smORFs that encode functionally important peptides is distinguishing them from the enormous number of smORFs present in the genome by chance (e.g. 260,000 in yeast) (*Basrai et al., 1997*). Many groups have identified and categorized smORFs with coding potential using signatures of evolutionary conservation, ribosomal profiling, and mass spectrometry (*Saghatelian and Couso, 2015*; *Couso and Patraquim, 2017*; *Plaza et al., 2017*). Together, these approaches suggest there may be hundreds, possibly thousands, of unannotated smORF genes. However, these 'omics methods do not tell us which smORFs encode peptides with important biological functions.

Functional characterization of smORF genes in cell lines and model organisms has the potential to confidently identify novel peptides. Historically, unbiased genetic screens and gene cloning led to the fortuitous identification and characterization of smORF peptides (e.g. POLARIS *Casson et al., 2002*, RpL41 *Suzuki et al., 1990*, Nedd4 *Kumar et al., 1993*, *Drosophila* pri/tal *Galindo et al., 2007*). More recently, candidate bioinformatically-predicted smORF-encoded peptides (aka SEPs) have been targeted for characterization (e.g. DWORF *Nelson et al., 2016*, Elabela/toddler *Chng et al., 2013*; *Pauli et al., 2014*, Myomixer *Bi et al., 2017*, Myoregulin *Anderson et al., 2015*, and Sarcolamban *Magny et al., 2013*, and Hemotin *Pueyo et al., 2016*). Collectively, these studies have been invaluable for assigning biological functions to smORF peptides. Therefore, continued functional characterization is needed to tackle the enormous number of predicted smORF peptides.

Here, through an effort to systematically characterize human-conserved smORF genes in *Drosophila* (*Bosch et al., 2022*), we identified two previously unstudied smORF peptides CG32736-PB and CG42308-PA that we named Sloth1 and Sloth2 based on their mutant phenotypes. Remarkably, both peptides are translated from the same transcript and share amino acid sequence similarity, suggesting that they encode paralogs. Loss-of-function analysis revealed that each peptide is essential for viability, and mutant animals exhibit defective neuronal function and photoreceptor degeneration. These phenotypes can be explained by our finding that Sloth1 and Sloth2 localize to mitochondria and play an important role in complex III assembly. Finally, we propose that both peptides bind in a shared complex. These studies uncover two new components of the mitochondria and demonstrate how functional characterization of smORFs will lead to novel biological insights.

## Results

### *sloth1* and *sloth2* are translated from the same transcript and are likely distantly related paralogs

Current gene annotations for *sloth1* and *sloth2* (aka *CG32736* and *CG42308*, respectively) indicate that they are expressed from the same transcript (Flybase, *Figure 1A*), known as a bicistronic (or dicistronic) gene (*Blumenthal, 2004*; *Crosby et al., 2015*; *Karginov et al., 2017*). For example, nearby transcription start sites (*Figure 1A*) are predicted to only generate a single transcript (*Hoskins et al., 2011*). In addition, a full-length transcript containing both smORFs is present in the cDNA clone RE60462 (GenBank Acc# AY113525), which was derived from an embryonic library (*Stapleton et al., 2002*), and we detected the full-length bicistronic transcript by RT-PCR amplification from total RNA from 3rd instar larvae, adult flies, and S2R+ cells (*Figure 1—figure supplement 1*). In addition, the encoded peptides Sloth1 and Sloth2 have subtle sequence similarity (27%), are similar in size (79aa and 61aa, respectively), and each contain a predicted single transmembrane domain (*Figure 1B*). While this type of gene structure is relatively rare in eukaryotes (*Blumenthal, 2004*; *Karginov et al., 2017*), there are known cases in *Drosophila* of multicistronic transcripts encoding smORF paralogs – the *pri/tal* locus (*Galindo et al., 2007*) and the *Sarcolamban* locus (*Magny et al., 2013*). Furthermore, it is well known that paralogs are often found adjacent to each other in the genome due to tandem duplication (*Taylor and Raes, 2004*). Therefore, we propose that *sloth1* and *sloth2* are paralogs translated from the same transcript.

Sloth1 and Sloth2 closely resemble their human orthologs (SMIM4 and C12orf73), based on sequence similarity, similar size, and presence of a transmembrane domain (*Figure 1B*). Like Sloth1

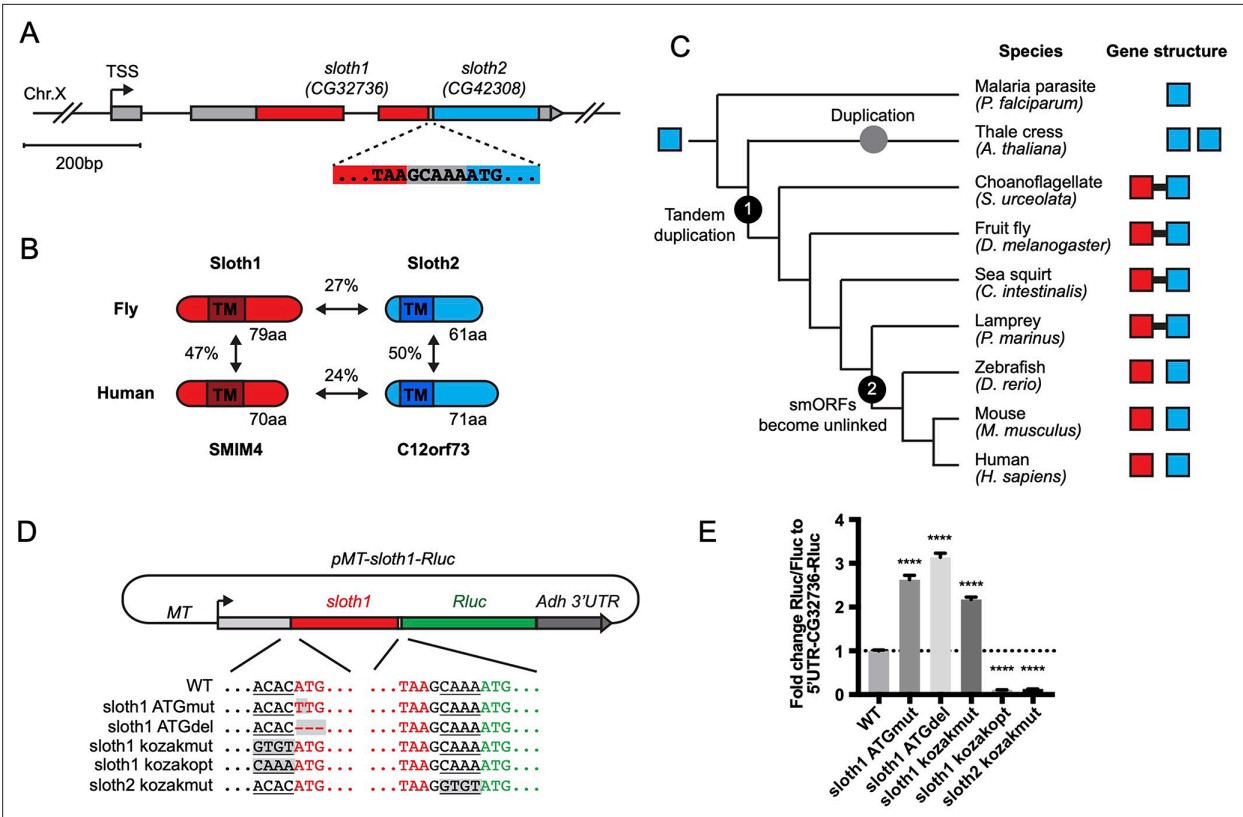

**Figure 1.** Bicistronic gene structure of the smORFs *sloth1* and *sloth2*. (**A**) Bicistronic gene model for *sloth1* and *sloth2*. Zoom in shows intervening sequence (*GCAAA*) between *sloth1* stop codon and *sloth2* start codon. (**B**) Comparison of protein structure, amino acid length size, and amino acid percent identity between *Drosophila* and Human orthologs. Shaded rectangle indicates predicted transmembrane (TM) domain. (**C**) Phylogenetic tree of *sloth1* and *sloth2* orthologs in representative eukaryotic species. Linked gene structure (candidate bicistronic transcript or adjacent separate transcripts) is indicated by a black line connecting red and blue squares. (**D**) Plasmid reporter structure of *pMT-sloth1-Rluc* and derivatives. Kozak sequences upstream of start codon are underlined. Mutations indicated with shaded grey box. pMT = Metallothionein promoter. RLuc = Renilla Luciferase. (**E**) Quantification of RLuc luminescence/Firefly Luciferase, normalized to *pMT-sloth1-Rluc*, for each construct. Significance of mutant plasmid luminescence was calculated with a T-Test comparing to *pMT-sloth1-Rluc*. Error bars are mean with SEM. **** p≤0.0001. N=4 biological replicates.

The online version of this article includes the following figure supplement(s) for figure 1:

**Figure supplement 1.** Related to *Figure 1*.

and Sloth2, SMIM4 and C12orf73 also have subtle amino acid sequence similarity to each other (*Figure 1B*). In addition, *sloth1* and *sloth2* are conserved in other eukaryotic species (*Figure 1C*). Remarkably, *sloth1* and *sloth2* orthologs in choanoflagelate, sea squirt, and lamprey exhibit a similar bicistronic gene architecture as *Drosophila* (*Figure 1C*, *Supplementary file 1*). In contrast, *sloth1* and *sloth2* orthologs in jawed vertebrates (e.g. mammals) are located on different chromosomes (e.g. human Chr.3 and Chr.12, respectively). Interestingly, we only found one ortholog similar to *sloth2* in the evolutionarily distant *Plasmodium*, and two orthologs similar to *sloth2* in *Arabidopsis*, which are located on different chromosomes (*Figure 1C*). Therefore, we hypothesize that the *sloth1* and *sloth2* ORFs duplicated from an ancient single common ancestor ORF and became unlinked in animals along the lineage to jawed vertebrates.

We next investigated *sloth1* and *sloth2* translation parameters and efficiency, since their ORFs are frameshifted relative to each other (*Figure 1A*) and they are not separated by an obvious internal ribosome entry site (IRES; *Van Der Kelen et al., 2009*). Remarkably, only five nucleotides separate the stop codon of the upstream ORF (*sloth1*) and the start codon of the downstream ORF (*sloth2*; *Figure 1A*). Therefore, *sloth1* should be translated first and inhibit translation of *sloth2*, similar to the functions of upstream ORFs (uORFs) (*Thompson, 2012*). However, *sloth1* has a non-optimal Kozak sequence 5' to the start codon (*ACAC*ATG) and *sloth2* has an optimal Kozak (*CAAA*ATG; *Cavener, 1987*). Therefore, scanning ribosomes may occasionally fail to initiate translation on *sloth1*, in which

case they would continue scanning and initiate translation on *sloth2*, known as 'leaky scanning' translation (***Thompson, 2012***).

To test this translation model, we constructed an expression plasmid with the *Renilla Luciferase* (*RLuc*) reporter gene downstream of *sloth1* (*sloth1-RLuc*), while retaining non-coding elements of the original transcript (5' UTR, Kozak sequences, 5 bp intervening sequence) (***Figure 1D***). By transfecting this reporter plasmid into *Drosophila* S2R+ cells, along with a *Firefly Luciferase* (*FLuc*) control plasmid, we could monitor changes in translation of the downstream ORF by the ratio of RLuc/FLuc luminescence. Using derivatives of the reporter plasmid with Kozak or ATG mutations, we found that translation of the downstream ORF increased when translation of *sloth1* was impaired (***Figure 1E***). Reciprocally, translation of the downstream ORF was decreased when *sloth1* translation was enhanced with an optimal Kozak. These results suggest that *sloth1* inhibits translation of *sloth2*, and that balanced translation of both smORFs from the same transcript might be achieved by suboptimal translation of *sloth1*.

## *sloth1* and *sloth2* are essential in *Drosophila* with non-redundant function

To determine if *sloth1* and *sloth2* have important functions in *Drosophila*, we used in vivo loss-of-function genetic tools. We used RNA interference (RNAi) to knock down the *sloth1-sloth2* bicistronic transcript. Ubiquitous expression of an shRNA targeting the *sloth1* coding sequence (***Figure 2A***) led to significant knockdown of the *sloth1-sloth2* transcript in 3rd instar larvae (***Figure 2B***), as determined by two different primer pairs that bind to either the *sloth1* or *sloth2* coding sequence. Ubiquitous RNAi knockdown of *sloth1-sloth2* throughout development led to reduced number of adult flies compared to a control (***Figure 2C***). This reduced viability was largely due to adult flies sticking in the food after they eclosed from their pupal cases (***Figure 2D***). Escaper knockdown flies were slow-moving and had 30% climbing ability compared to control flies (***Figure 2E***). RNAi knockdown flies also had short scutellar bristles (***Figure 2F***).

We confirmed our RNAi results using CRISPR/Cas9 to generate somatic knockout (KO) flies. By crossing flies ubiquitously expressing Cas9 (*Act-Cas9*) with flies expressing an sgRNA that targets the coding sequence of either *sloth1* or *sloth2* (***Figure 2A***, ***Figure 2—figure supplement 1A***), the resulting progeny will be mosaic for insertions and deletions (indels) that cause loss of function in somatic cells (***Port et al., 2014***; ***Xue et al., 2014***). Both *sloth1* and *sloth2* somatic KO flies had significantly reduced viability compared to controls (***Figure 2G***). Furthermore, escaper adults had short scutellar bristles (***Figure 2H***) and frequently appeared sluggish. Importantly, similar phenotypes were observed when targeting either *sloth1* or *sloth2*.

Next, we further confirmed our loss of function results using CRISPR/Cas9 in the germ line to generate KO lines for *sloth1* and *sloth2*. These reagents are particularly important to test if *sloth1* and *sloth2* have redundant function by comparing the phenotypes of single and double null mutants. We generated four KO lines (***Figure 2A***, ***Figure 2—figure supplement 1A-C***): (1) a frameshift indel in *sloth1* (*sloth1*-KO), (2) a frameshift indel in *sloth2* (*sloth2*-KO), (3) a 552 bp deletion of the *sloth1* and *sloth2* reading frames (dKO), and (4) a knock-in of the reporter gene *Gal4* that removes *sloth1* and *sloth2* coding sequences (*Gal4-KI*). Since *sloth1* and *sloth2* are on the X-chromosome, we analyzed mutant hemizygous male flies. All four mutant lines were hemizygous lethal, which were rescued by a genomic transgene (***Figure 2I***), ruling out off-target lethal mutations on the X-chromosome. Like RNAi and somatic KO results, rare mutant adult escaper flies had slower motor activity (***Figure 2J***) and short scutellar bristles (***Figure 2K***). Furthermore, the short scutellar bristle phenotype and slower motor activity was rescued by a genomic transgene (***Figure 2J and K***).

The phenotypic similarity of single and double mutants suggests that *sloth1* and *sloth2* are not functionally redundant. However, since both ORFs are encoded on the same transcript, it is unclear if mutating one ORF will affect the other. For example, a premature stop codon can induce nonsense mediated decay of an entire transcript (***Nickless et al., 2017***). To address this possibility, we performed additional fly lethality rescue experiments. First, transheterozygous female flies (*sloth1-KO/+*, *sloth2-KO/+*) were viable and had normal scutellar bristles. Second, we created single ORF versions of a genomic rescue transgene – {Δ*sloth1-sloth2*} and {*sloth1-*Δ*sloth2*} (***Figure 2—figure supplement 1A***). We found that *sloth1-KO* lethality could only be rescued by {*sloth1-*Δ*sloth2*}, and vice versa, *sloth2-KO* lethality could only rescued by {Δ*sloth1-sloth2*} (***Figure 2L***). Furthermore, single

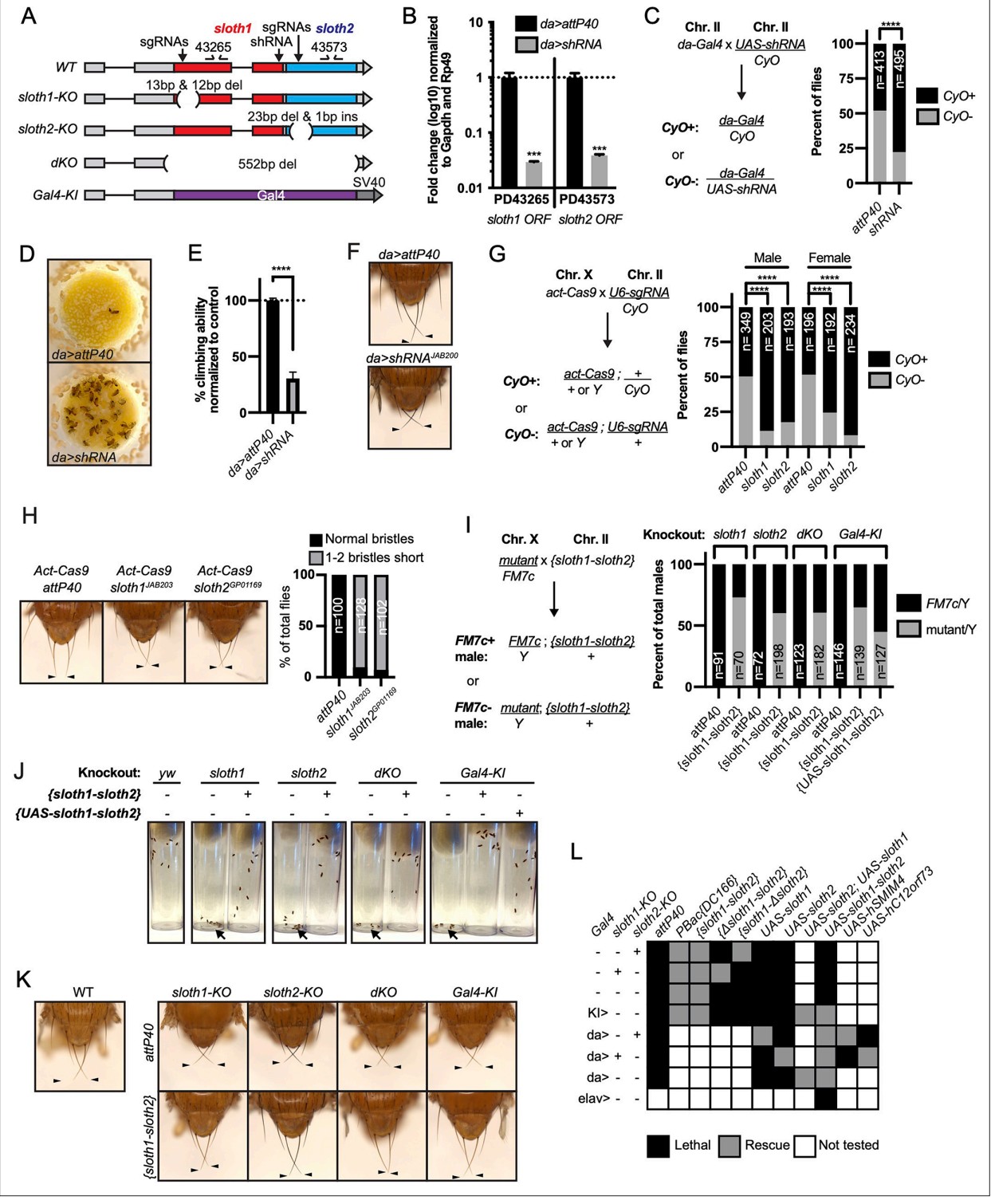

**Figure 2.** *sloth1* and *sloth2* loss of function analysis. (**A**) *sloth1-sloth2* transcript structure with shRNA and sgRNA target locations, primer binding sites, in/del locations, and knock-in Gal4 transgene. (**B**) qPCR quantification of RNAi knockdown of the *sloth1-sloth2* transcript. Significance of fold change knockdown was calculated with a T-Test comparing to *da>attP40* for PD43265 and PD43573. Error bars show mean with SEM. p-values *** p≤0.001. N=6. (**C**) Quantification of adult fly viability from *sloth1-sloth2* RNAi knockdown. Fly cross schematic (left) and graph (right) with percentage of progeny with or without the CyO balancer. Ratios of balancer to non-balancer were analyzed by Chi square test, **** p≤0.0001. Sample size (**N**) indicated on graph. (**D**) Pictures of fly food vials, focused on the surface of the food. *da>shRNA* flies are frequently found stuck in the fly food. (**E**) Quantification of adult fly climbing ability after *sloth1* and *sloth2* RNAi. Significance calculated with a T-test, **** p≤0.0001. Error bars show mean with SD. N=3 biological

*Figure 2 continued*

replicates. (**F**) Stereo microscope images of adult fly thorax to visualize the scutellar bristles. RNAi knockdown by *da-Gal4* crossed with either *attP40* or *UAS-shRNA*[JAB200]. Arrowheads point to the two longest scutellar bristles. (**G**) Quantification of adult fly viability from *sloth1-sloth2* somatic knockout. Fly cross schematic (left) and graph (right) with percentage of progeny with or without the CyO balancer. Ratios of balancer to non-balancer were analyzed by Chi square test, **** p≤0.0001. Sample size (**N**) indicated on graph. (**H**) (Left) Stereo microscope images of adult fly thorax to visualize the scutellar bristles. Somatic knockout performed by crossing *Act-Cas9* to sgRNAs. (Right) Quantification of the frequency of adult flies with at least one short scutellar bristle after somatic KO of *sloth1* or *sloth2*. Sample sizes indicated on graph. Arrowheads point to the two longest scutellar bristles. (**I**) Quantification of adult fly viability from *sloth1-sloth2* hemizygous knockout in males and rescue with a genomic transgene or *UAS-sloth1-sloth2* transgene. Fly cross schematic (left) and graph (right) with percentage of male progeny with or without the FM7c balancer. Sample size (**N**) indicated on graph. (**J**) Still images from video of adult flies inside plastic vials. Images are 5 s after vials were tapped. Adult flies climb upward immediately after tapping. All flies are males. Each vial contains 10 flies, except dKO, which contains five flies. (**K**) Stereo microscope images of adult male fly thorax to visualize the scutellar bristles. *attP40* is used as a negative control. Arrowheads point to the two longest scutellar bristles. (**L**) Hemizygous mutant male genetic rescue experiments.

The online version of this article includes the following figure supplement(s) for figure 2:

**Figure supplement 1.** Related to *Figure 2*.

ORF rescue transgenes were unable to rescue the lethality of *dKO* and *Gal4-KI* lines (*Figure 2L*). Third, we used the Gal4/UAS system (*Brand and Perrimon, 1993*) to rescue mutant lethality with ubiquitously expressed cDNA transgenes. These results showed that single ORF KOs could only be rescued by expression of the same ORF (*Figure 2L*). Similar results were found by expressing cDNAs encoding the human orthologs (*Figure 2L*). In all, these results show that both *sloth1* and *sloth2* are essential, have similar loss of function phenotypes, are not functionally redundant with one another, and are likely to retain the same function as their human orthologs.

## Loss of *sloth1* and *sloth2* leads to defective neuronal function and degeneration

Since loss of *sloth1* and *sloth2* caused reduced adult mobility and climbing defects (*Figure 2E and J*), we speculated that the two peptides normally play an important role in the brain or muscle. To determine where *sloth1* and *sloth2* are expressed, we used the *Gal4-KI* line as an in vivo transcriptional reporter. *Gal4-KI* mobility defects and lethality could be rescued by expressing the entire bicistronic transcript (*UAS-sloth1-sloth2*; *Figure 2J and L*), or coexpression of both smORFs as cDNA (*UAS-sloth1* and *UAS-sloth2*; *Figure 2L*). Thus, the *Gal4-KI* line is likely an accurate reporter of *sloth1* and *sloth2* expression. By crossing *Gal4-KI* flies with a *UAS-GFP* fluorescent reporter, we observed strong GFP expression in larval (*Figure 3A and B*) and adult brains (*Figure 3C*). In addition, *Gal4-KI* is expressed in motor neurons at the larval neuromuscular junction (NMJ) (*Figure 3D*) and in larval brain cells that are positive for the neuronal marker Elav (*Figure 3E*).

We then tested if *sloth1* and *sloth2* were important for neuronal function by measuring neuronal electrical activity in *dKO* animals. Electrical recordings taken from the larval NMJ showed that *dKO* motor neurons have normal excitatory junction potential (EJP) under resting conditions at 0.75 mM $Ca^{2+}$ (*Figure 4—figure supplement 1*). However, under high frequency stimulation (10 hz), *dKO* NMJs could not sustain a proper response (*Figure 4A*), indicating that there is a defect in maintaining synaptic vesicle pools. Importantly, this phenotype is rescued by a genomic transgene. To test if a similar defect is present in the adults, we assessed phototransduction and synaptic transmission in photoreceptors via electroretinogram (ERG) recordings (*Wu and Wong, 1977*; *Hardie and Raghu, 2001*). ERGs recorded from young (1–3 days old) *dKO* photoreceptors showed an amplitude similar to that of genomic rescue animals (*Figure 4B*). However, upon repetitive light stimulation, ERG amplitudes were significantly reduced (*Figure 4B*), suggesting a gradual loss of depolarization. Similar results were observed when young flies were raised in 24 hr dark (*Figure 4C*). Moreover, ERG traces also showed a progressive loss of 'on' and 'off' transients (*Figure 4B and C*), which is indicative of decreased synaptic communication between the photoreceptor and the postsynaptic neurons. ERG phenotypes are rescued by a full-length genomic rescue transgene, but not by single ORF rescue transgenes (*Figure 4B and C*). To test if loss of both *sloth1* and *sloth2* lead to neurodegeneration, we aged the animals for 4 weeks in 12 hr light/dark cycle or constant darkness and recorded ERGs. Similar to young animals, aged animals raised in light/dark conditions also displayed a reduction in ERG amplitude upon repetitive stimulation (*Figure 4E*). These results indicate that both *sloth1* and *sloth2*

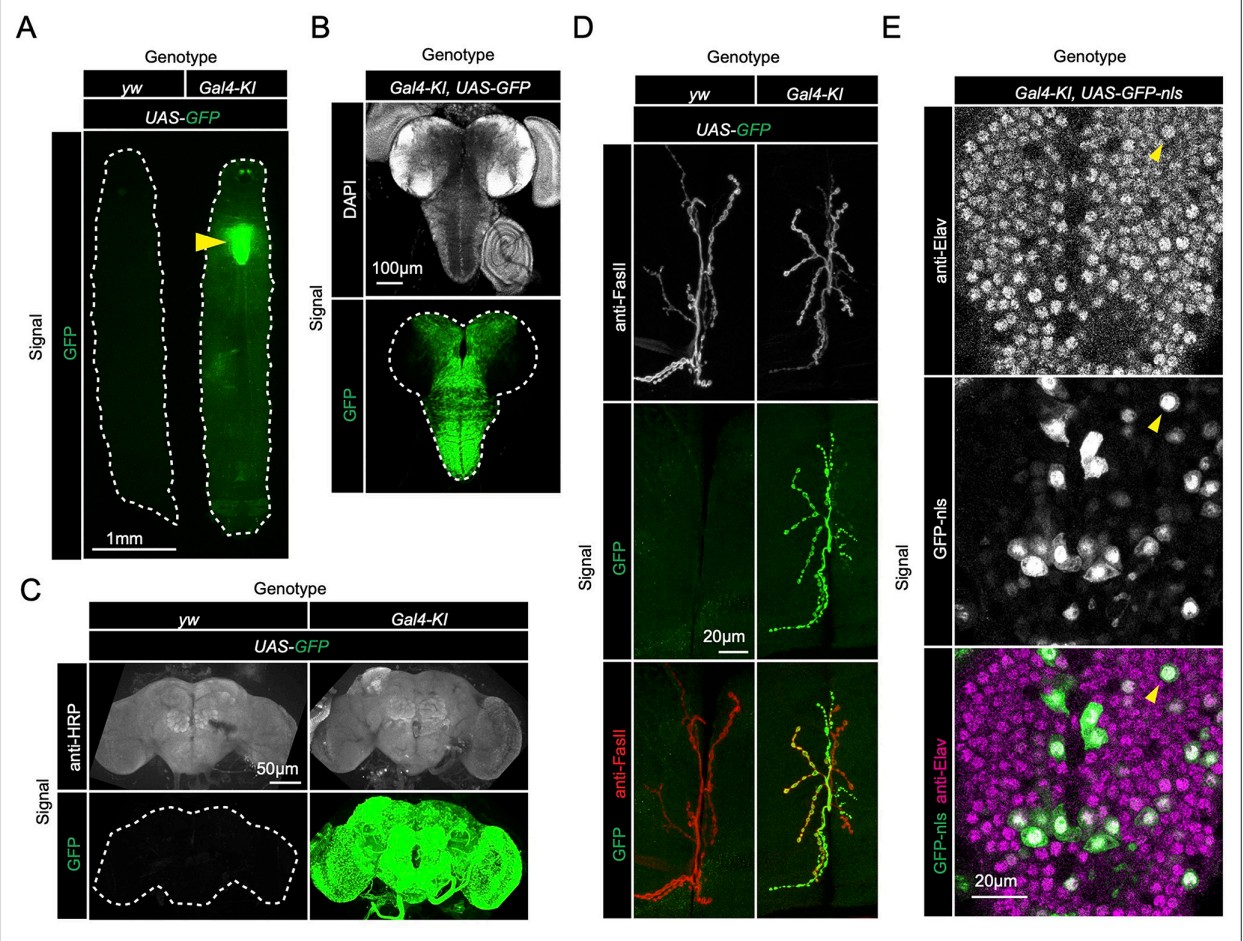

**Figure 3.** *sloth1-sloth2* are expressed in neurons. (**A**) Fluorescent stereo microscope images of 3rd instar larvae expressing GFP with indicated genotypes. (**B**) Fluorescent compound microscope image of 3rd instar larval brain expressing *UAS-GFP*. DAPI staining labels nuclei. (**C**) Confocal microscopy of adult brain with indicated genotypes. Anti-HRP staining labels neurons. (**D**) Confocal microscopy of the 3rd instar larval NMJ at muscle 6/7 segment A2 expressing *UAS-GFP*. Anti-FasII staining labels the entire NMJ. (**E**) Confocal microscopy of the 3rd instar larval ventral nerve cord (VNC) expressing *Gal4-KI, UAS-GFP-nls*. GFP-nls is localized to nuclei. Anti-Elav stains nuclei of neurons. Arrow indicates example nuclei that expresses UAS-GFP and is positive for Elav.

are required for sustained neuronal firing in larval motor neurons and adult photoreceptors. Interestingly, similar mutant phenotypes in the NMJ and photoreceptors are known to be due to defects in ATP production (***Verstreken et al., 2005***; ***Sandoval et al., 2014***; ***Jaiswal et al., 2015***).

In addition to measuring neuronal activity, we analyzed *dKO* neurons for changes in morphology and molecular markers. Confocal imaging of the NMJ in *dKO* 3rd instar larvae did not reveal obvious changes in synapse morphology or markers of synapse function (***Figure 5—figure supplement 1***). In contrast, using transmission electron microscopy (TEM) of sectioned adult eyes, we observed reduced photoreceptor number and aberrant morphology such as enlarged photoreceptors and thinner glia in *dKO* animals (***Figure 5A–C***), suggestive of degeneration. These phenotypes were rescued by a genomic transgene, but not with single ORF rescue constructs (***Figure 5A–C***, ***Figure 5—figure supplement 2***). Furthermore, these phenotypes were similar between young and aged flies, as well as aged flies raised in the dark (***Figure 5A–C***, ***Figure 5—figure supplement 2***). It is known that mutations affecting the turnover of Rhodopsin protein (Rh1) can lead to photoreceptor degeneration (***Alloway et al., 2000***; ***Jaiswal et al., 2015***). To test if this mechanism is occurring in *dKO* photoreceptors, we imaged Rh1 protein levels using confocal microscopy. We observed Rh1 accumulation in degenerating *dKO* photoreceptors in 4-week aged flies exposed to light (***Figure 5D***). However, Rh1 accumulation was milder in 4-week aged flies raised in the dark (***Figure 5—figure supplement 3***).

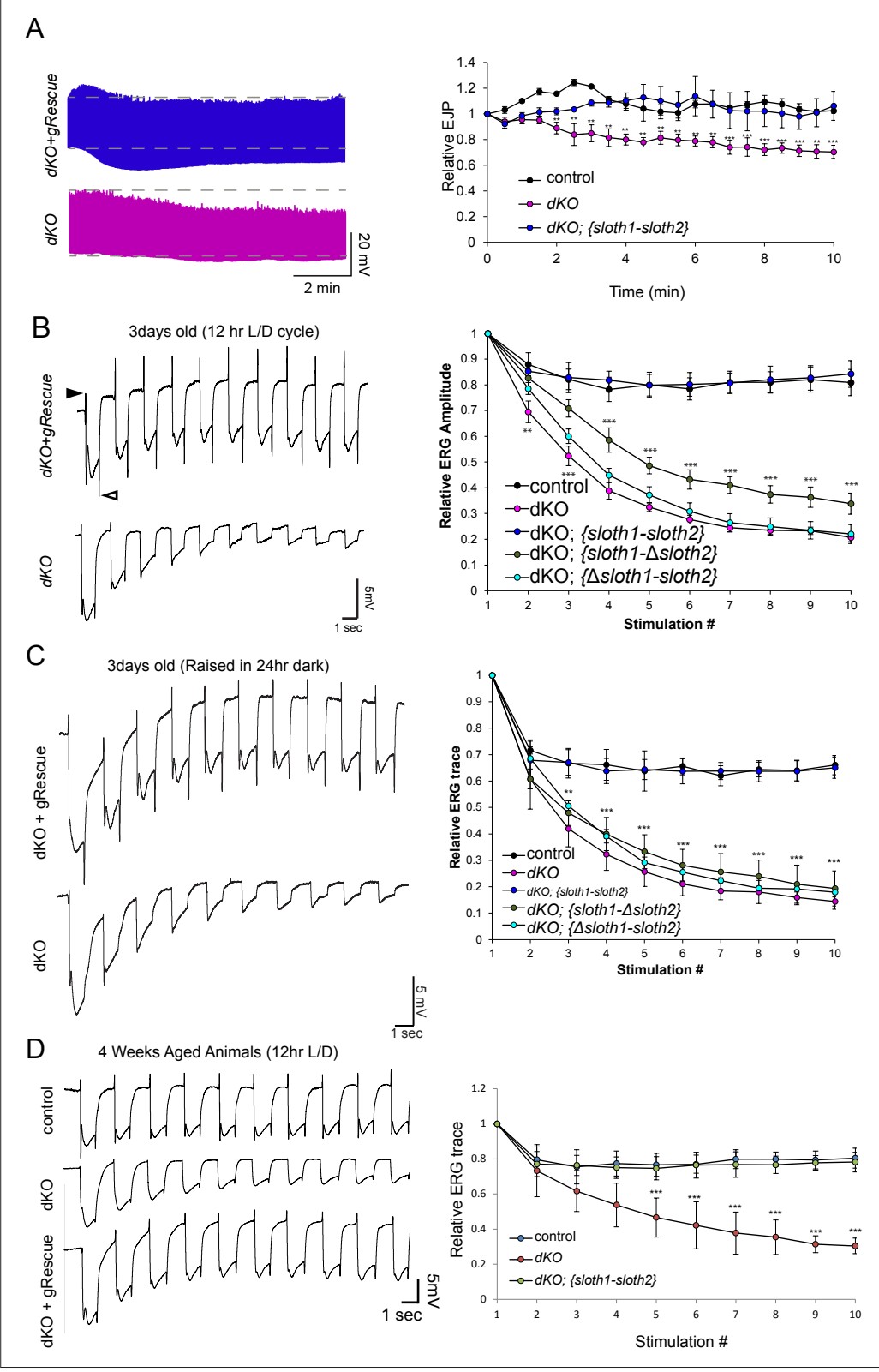

**Figure 4.** *sloth1-sloth2* are important for neuronal function. (**A**) Traces of electrical recordings from 3rd instar larval NMJ in control, *dKO*, and *dKO + genomic rescue* animals over 10 min under high-frequency stimulation (10 Hz). Graph on right is a quantification of the relative excitatory junction potential (EJP) for indicated genotypes. Error bars show mean with SD. N≥5 larvae per genotype. Significance for each genotype was calculated with a T-Test

*Figure 4 continued on next page*

*Figure 4 continued*

comparing to control flies. (**B–D**) Traces of electroretinogram (ERG) recordings from adult eye photoreceptors upon repetitive stimulation with light (left) and quantification of the relative ERG amplitude for indicated genotypes (right). Error bars show mean with SD. N≥6 larvae per genotype. ** p≤0.01, *** p≤0.001. Significance for each genotype was calculated with a T-Test comparing to control flies. (**B**) Recordings were taken from 1 to 3 days post-eclosion animals that were raised in a 12 hr light/dark cycle. 'On' and 'Off' transients indicated by closed and open arrowhead, respectively. (**C**) Recordings were taken from 1 to 3 days post-eclosion animals that were raised in a 24 hr dark. (**D**) Recordings were taken from four week aged animals that were raised in a 12 hr light/dark cycle.

The online version of this article includes the following figure supplement(s) for figure 4:

**Figure supplement 1.** Related to *Figure 4*.

These results point out that light stimulation, and hence activity, enhance degeneration due to Rh1 accumulation in *dKO* animals.

## Sloth1 and Sloth2 localize to mitochondria and their loss impairs normal respiration and ATP production

Mitochondrial dysfunction in *Drosophila* is known to cause phenotypes that are reminiscent of loss of *sloth1* and *sloth2*, such as pupal lethality, reduced neuronal activity, photoreceptor degeneration, and Rh1 accumulation in photoreceptors (*Jaiswal et al., 2015*). Therefore, we investigated the possible role of Sloth1 and Sloth2 in mitochondria.

Prior to our work, a large-scale study of human protein localization suggested that SMIM4 and C12orf73 localize to mitochondria in cultured cells (*Thul et al., 2017*). SMIM4 has a predicted mitochondrial targeting sequence using MitoFates (*Fukasawa et al., 2015*) (0.842), but C12orf73, Sloth1, and Sloth2 do not (0.0016, 0.016, 0.009, respectively). In addition, SMIM4 and Sloth1 are predicted to localize to the mitochondrial inner membrane using DeepMito (0.93 and 0.73, respectively), but C12orf73 and Sloth2 are not (0.66 and 0.49, respectively) (*Savojardo et al., 2020*). To test if Sloth1/2 localize to mitochondria in *Drosophila*, we transfected S2R+ cells with Sloth1-FLAG or Sloth2-FLAG. Both Sloth1 and Sloth2 proteins colocalized with the mitochondrial marker ATP5α (*Figure 6A*). Furthermore, Sloth1-FLAG and Sloth2-FLAG were enriched in mitochondrial fractions relative to cytoplasmic fractions (*Figure 6B*). Similar results were observed using stable S2R+ cell lines that express streptavidin binding peptide (SBP) tagged Sloth1 or Sloth2 under a copper inducible promoter (*MT-Sloth1-SBP* and *MT-Sloth2-SBP*) (*Figure 6C*).

Next, we raised antibodies to Sloth1/2 to determine their endogenous localization. Using two independently generated antibodies for each peptide, immunolocalization in larval brains from wild-type or *sloth1/2* dKO animals showed no overlapping signal with a mitochondrial marker and no clear signal above background (*Figure 6—figure supplement 1*). Furthermore, we did not detect Sloth1 or Sloth2 bands of the expected molecular weight on western blots from wild-type S2R+ whole cell lysates or isolated mitochondria using anti-Sloth1, anti-Sloth2, anti-SMIM4, or anti-C12orf73 (*Figure 6—figure supplement 2A-C*). In contrast, anti-Sloth1 western blots of mitochondria isolated from 3rd instar larvae and adult thoraxes showed a<15 kDa band that is absent from *sloth1/2* KO or RNAi samples (*Figure 6—figure supplement 2D*), suggesting this band corresponds to endogenous Sloth1. Unfortunately, anti-Sloth2 failed to detect a similar band under the same conditions (*Figure 6—figure supplement 2D*).

Since our Sloth1/2 antibodies may not be sensitive enough to detect the endogenous peptides, we generated a stable S2R+ cell line expressing *sloth1/2* transcript under a copper inducible promoter (*MT-sloth1/2*) and induced expression for 16 hr. Anti-Sloth1 and anti-Sloth2 western blots of mitochondria isolated from *MT-sloth1/2* cells detected <15 kDa bands that did not appear in wild-type S2R+ cells, and thus are likely Sloth1 and Sloth2 peptides translated from the overexpressed *sloth1/2* transcript (*Figure 6—figure supplement 2B*). Furthermore, Sloth1 and Sloth2 were enriched in *MT-sloth1/2* mitochondrial fractions relative to cytoplasmic fractions (*Figure 6D*), similar to the results obtained with FLAG and SBP-tagged peptides (*Figure 6B–C*). Based on their amino acid sequence, Sloth1 and Sloth2 are predicted to run at 9.3 kDa and 6.7 kDa, respectively. While Sloth1 does appear to run larger than Sloth2, both peptides run ~2 kDa larger than expected (*Figure 6D*).

A method of assaying defects in mitochondrial function is measuring cellular oxygen consumption from live cells with a Seahorse stress test. Since this typically involves assaying a monolayer of cells, we

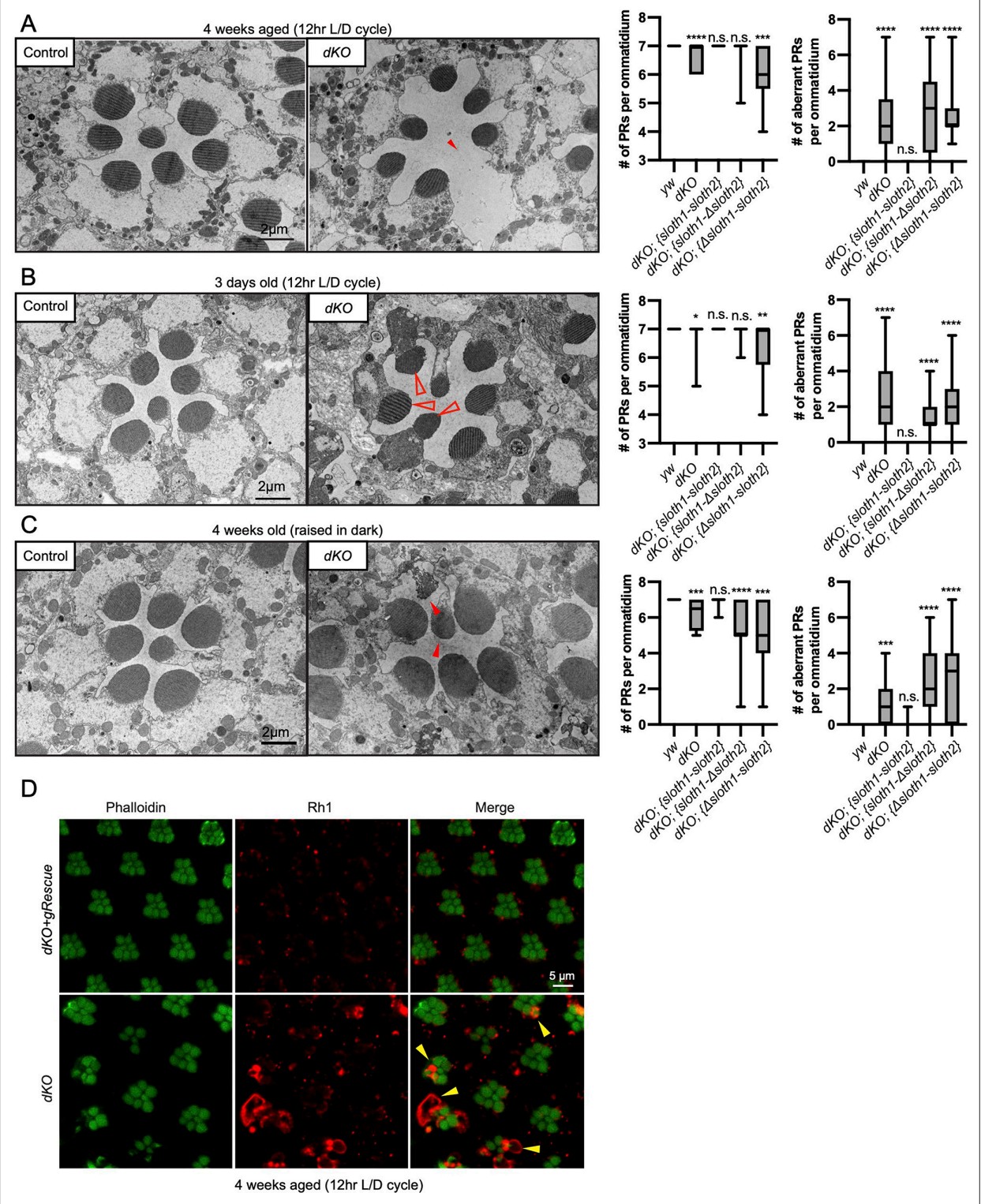

**Figure 5.** Loss of *sloth1-sloth2* causes neurodegeneration. (A–C) Transmission electron microscopy (TEM) images of sectioned adult eye photoreceptors (left) and quantification of photoreceptor number and aberrant photoreceptors (right). Scalebar is 2 µm. Filled red arrows indicate dead or dying photoreceptors. Open red arrows indicate unhealthy photoreceptors. Error bars show mean with SD. N≥8 ommatidium per genotype. (A) Four weeks old raised in a 12 hr light/dark cycle. (B) 3 days old raised in a 12 hr light/dark cycle. (C) Four weeks old raised in 24 hr dark. (D) Confocal microscopy of adult eye photoreceptors stained with phalloidin (green) and anti-Rh1 (red). Animals were 4 weeks old and raised in a 12 hr light/dark cycle. Arrowheads indicate photoreceptors with higher levels of Rh1.

*Figure 5 continued on next page*

Figure 5 continued

The online version of this article includes the following figure supplement(s) for figure 5:

**Figure supplement 1.** Related to *Figure 5*.

**Figure supplement 2.** Related to *Figure 5*.

**Figure supplement 3.** Related to *Figure 5*.

generated KO S2R+ cell lines using CRISPR/Cas9. Compared to control cells, single KO and double KO S2R+ cells (*Figure 7—figure supplement 1A, B*) had reduced basal respiration (*Figure 7A and B*), ATP production (*Figure 7—figure supplement 1C*), and proton leaks (*Figure 7—figure supplement 1D*). Results were similar for single KO and dKO lines. These results suggest that both *sloth1* and *sloth2* are required to support normal mitochondrial respiration in S2R+ cells.

Next, we assayed *sloth1* and *sloth2* mutant flies for defects in mitochondrial function. ATP levels are an important indicator of mitochondrial function (*Kann and Kovács, 2007*; *Golpich et al., 2017*) and mutations in *Drosophila* mitochondrial genes can lead to reduced ATP levels (*Jaiswal et al., 2015*). Indeed, *dKO* larvae had ~60% ATP compared to control larvae, which was rescued by a genomic transgene (*Figure 7C*). Impaired mitochondrial function can also lead to cellular stress responses, such as increased expression of the mitochondrial chaperone Hsp60 (*Pellegrino et al., 2013*). Western blot analysis showed that *Drosophila* Hsp60 was elevated in lysates from mutant larval brains compared to control, and this effect was rescued by a genomic transgene (*Figure 7D*). Finally, mitochondrial dysfunction can cause changes in mitochondrial morphology and number (*Trevisan et al.,*

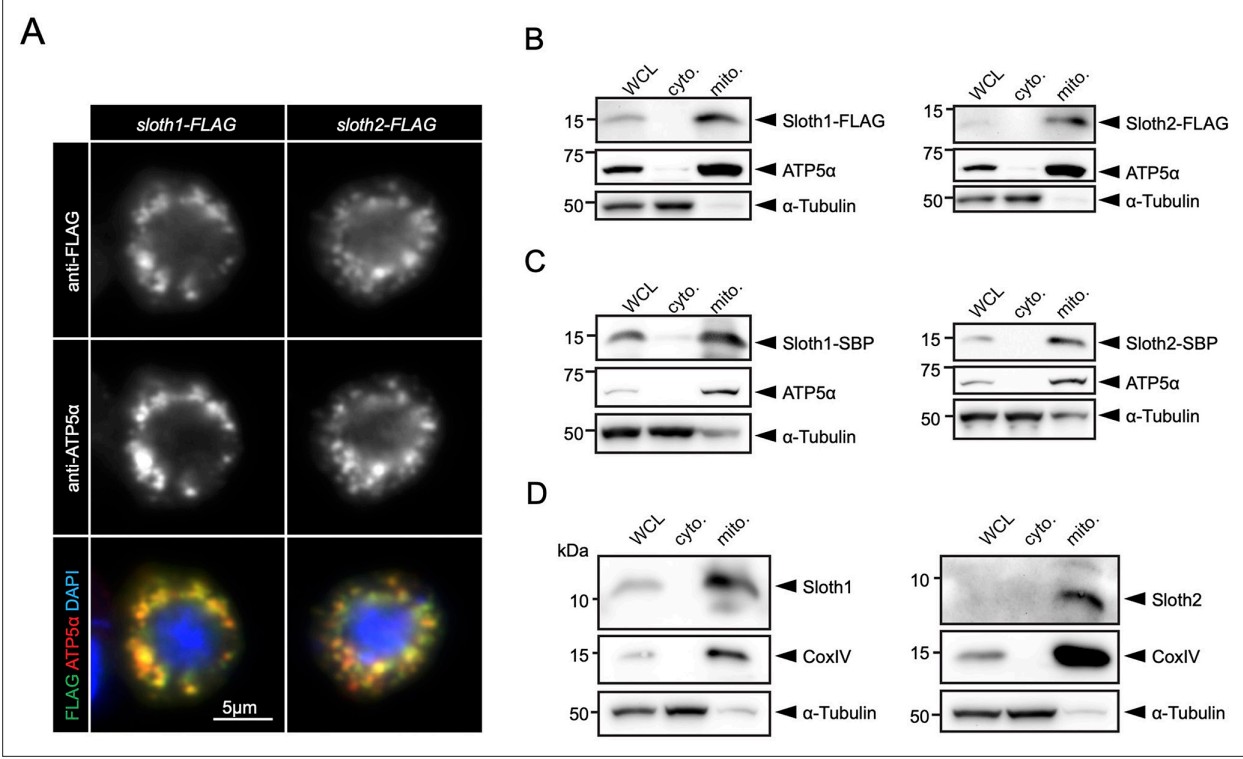

**Figure 6.** Sloth1 and Sloth2 localize to mitochondria. (**A**) Confocal microscopy of S2R+ cells transfected with Sloth1-FLAG or Sloth2-FLAG and stained with anti-FLAG (green) and anti-ATP5alpha (red). DAPI (blue) stains nuclei. (**B–D**) SDS-PAGE and western blotting of S2R+ cellular fractions. WCL = Whole Cell Lysate, cyto.=cytoplasmic lysate, mito.=mitochondrial lysate. Mitochondrial control = ATP5alpha, cytoplasmic control = alpha-tubulin. Each lane loaded equal amounts of protein (15 µg/lane). Blots were stripped and reprobed after detection of each antigen. (**B**) Transfected Sloth1-FLAG or Sloth2-FLAG. (**C**) Stable cells expressing copper-inducible Sloth1-SBP or Sloth2-SBP. (**D**) Stable cells expressing copper-inducible Sloth1-SBP or Sloth2-SBP.

The online version of this article includes the following figure supplement(s) for figure 6:

**Figure supplement 1.** Related to *Figure 6*.

**Figure supplement 2.** Related to *Figure 6*.

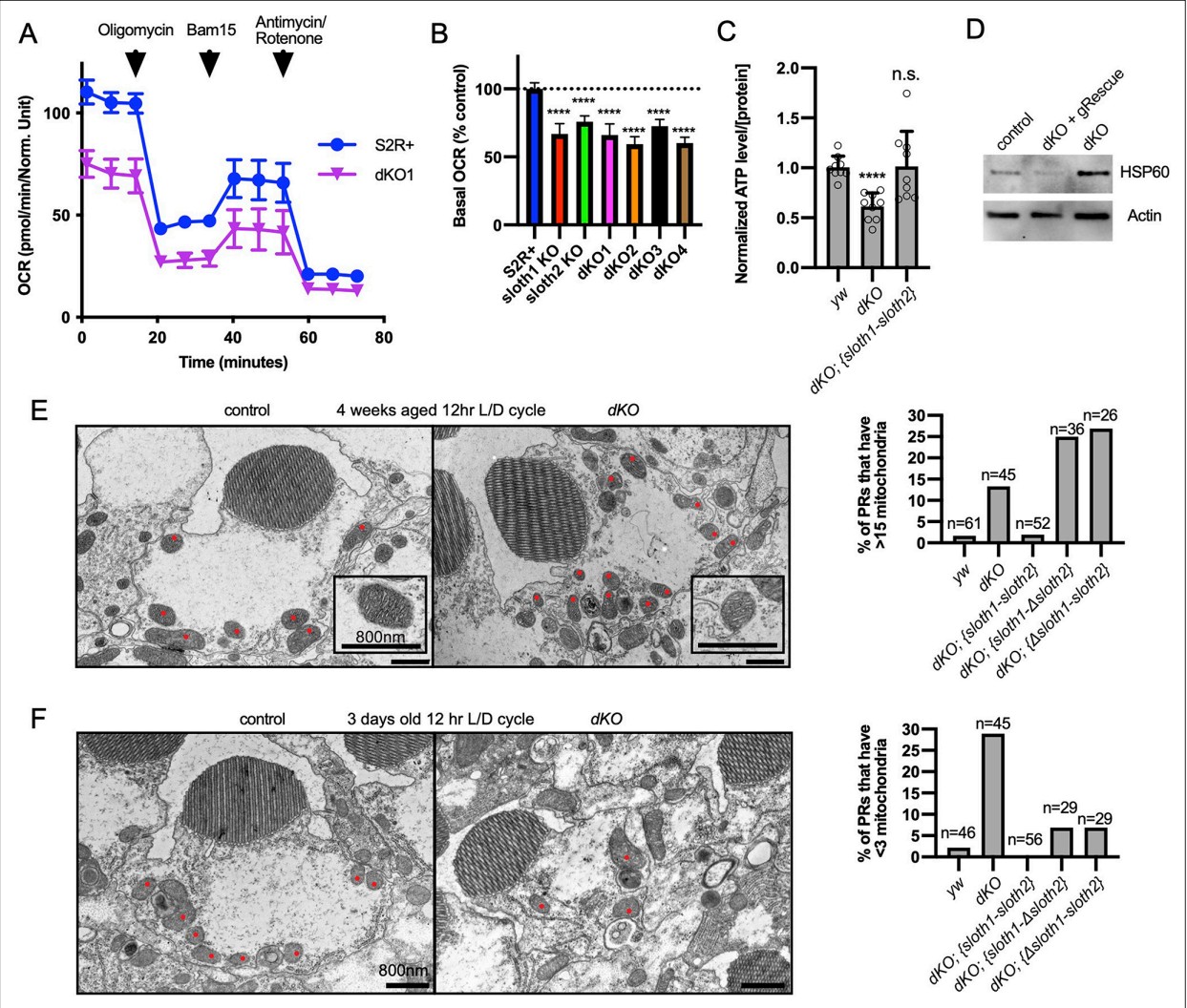

**Figure 7.** *sloth1-sloth2* are important for mitochondrial function. (**A**) Seahorse mitochondrial stress report for wildtype S2R+and dKO #1 cells. Error bars show mean with SD. N=6 for each genotype. (**B**) Quantification of basal OCR (timepoint 3) in panel A and including data from single KO and additional dKO cell lines. Significance of KO lines was calculated with a T-test compared to S2R+. Error bars show mean with SD. **** p≤0.0001. N=6 for each genotype. (**C**) Quantification of ATP levels in 3rd instar larvae. Error bars show mean with SEM. N=3 experiments. (**D**) Western blot from lysates of 3rd instar larval brains. (**E–F**) TEM images of sectioned adult photoreceptors (left) and quantification of mitochondria number (right). Mitochondria are indicated with red dots. Error bars show mean with SD. Sample size indicated on graph. (**E**) Adult flies are 4 weeks old and raised in a 12 hr light/dark cycle. (**F**) Adult flies are 3 days old and raised in a 12 hr light/dark cycle.

The online version of this article includes the following figure supplement(s) for figure 7:

**Figure supplement 1.** Related to *Figure 7*.

**Figure supplement 2.** Related to *Figure 7*.

*2018*). There were no obvious changes in mitochondrial morphology in mutant larval motor neurons (*Figure 5—figure supplement 1*, *Figure 7—figure supplement 1E*), and adult mutant photoreceptors contained mitochondria with normal cristae (*Figure 7E*). In contrast, mitochondrial number was increased in mutant photoreceptors in aged animals (*Figure 7E*, *Figure 7—figure supplement 2A*) and decreased in mutant photoreceptors in young animals (*Figure 7F*, *Figure 7—figure supplement 2B*). In all, these data suggest that Sloth1 and Sloth2 localize to mitochondria and are important to support respiration and ATP production.

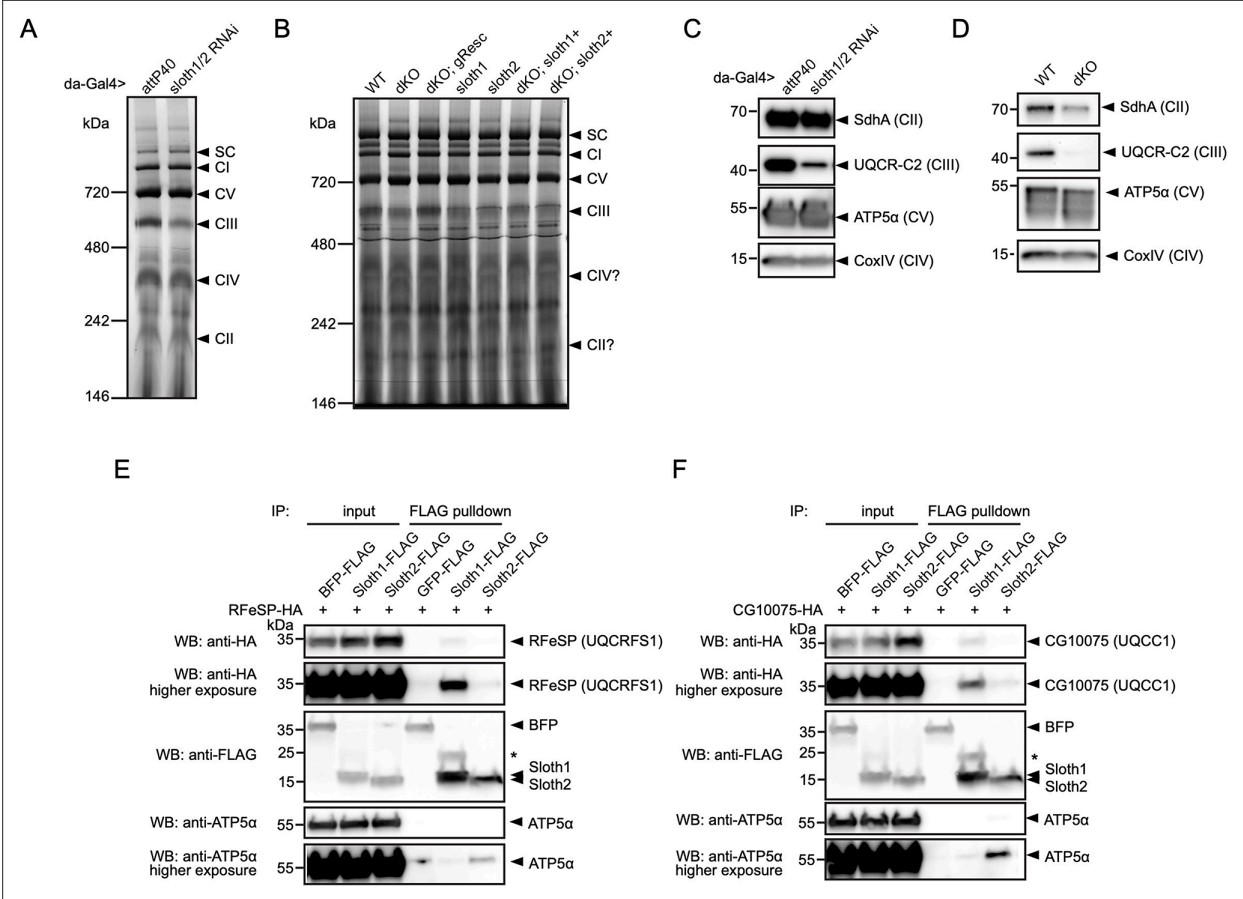

**Figure 8.** Sloth1 and Sloth2 physically interact with complex III and regulate its assembly. (**A–B**) Blue native PAGE gel of mitochondria isolated from (**A**) 10 adult thoraxes and (**B**) 10 whole 3rd instar larvae of indicated genotype. Bands corresponding to native respiratory complexes are indicated with arrowheads. (**C–D**) SDS-PAGE and western blotting of mitochondria isolated from (**C**) adult thorax and (**D**) whole 3rd instar larvae of indicated genotype. Each lane loaded equal amount of protein (15 μg). Blots were stripped and reprobed after detection of each antigen. (**E–F**) Western blots from co-immunoprecipitation experiments in transfected S2R+ cells using Sloth1-FLAG and Sloth2-FLAG as bait and either (**E**) RFeSP-HA or (**F**) CG10075-HA as prey. Blots were striped and reprobed after detection of each antigen. Arrowheads indicated expected band, asterisks indicate unknown bands.

## Sloth1/2 regulate respiratory complex III assembly

While our study was in preparation, two studies demonstrated that human SMIM4 and C12orf73 are inner mitochondrial membrane peptides important for complex III assembly and physically interact with complex III subunits (*Zhang et al., 2020*; *Dennerlein et al., 2021*). If Sloth1 or Sloth2 have similar roles in *Drosophila*, this could explain why *sloth1/2* mutant flies have reduced ATP production.

To test for a role in Sloth1/2 in respiratory complex assembly, we visualized the relative abundance of individual complexes and subunits in wild-type vs *sloth1/2* loss of function animals. First, we resolved native respiratory complexes using blue native polyacrylamide gel electrophoresis (BN-PAGE). Using mitochondria isolated from adult thorax, we identified the five respiratory complexes (CI, CII, CIII, CIV, CV) based on molecular weight and a previous study that established this protocol (*Garcia et al., 2017*). Importantly, a ~600 kDa band corresponding to complex III was diminished in mitochondria isolated from thoraxes with *sloth1/2* knockdown (*Figure 8A*). Similarly, the complex III band was diminished in mitochondria isolated from *sloth1/2* knockout 3rd instar larvae (*Figure 8B*). This change was rescued by a wild-type genomic transgene, but not single paralog transgenes (*Figure 8B*). Next, we detected individual respiratory subunits by SDS-PAGE and western blotting of isolated mitochondria. Using antibodies that recognize UQCR-C2, the fly homolog of human complex III subunit UQCRC2, we found that the ~40 kDa band corresponding to UQCR-C2 was diminished in mitochondria isolated from *sloth1/2* RNAi adult thoraxes (*Figure 8C*), as well as *sloth1/2* knockout 3rd instar larvae (*Figure 8D*).

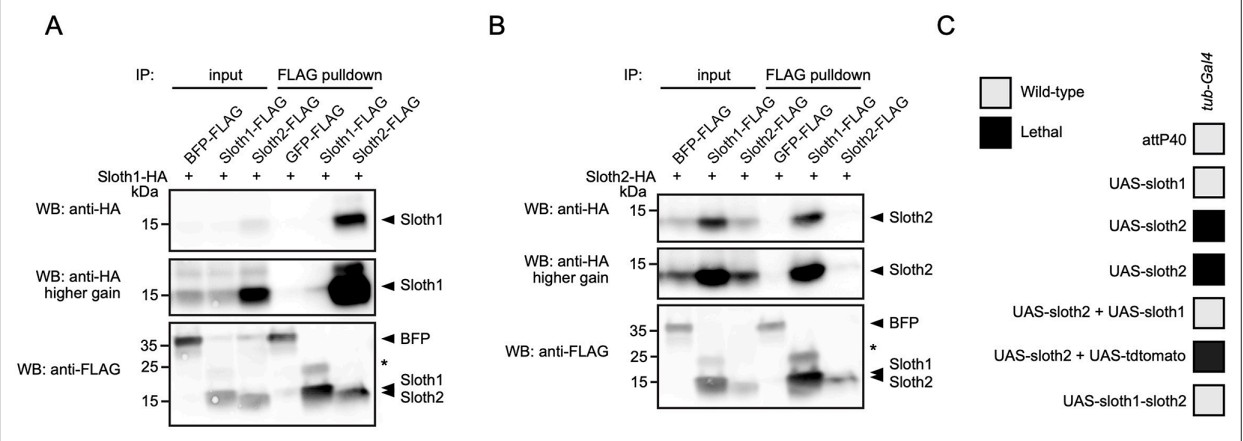

**Figure 9.** Sloth1 and Sloth2 act in a stoichiometric complex. (**A–B**) Western blots from co-immunoprecipitation experiments in transfected S2R+ cells. (**A–B**) Immunoprecipitation using Sloth1-FLAG and Sloth2-FLAG as bait and either (**A**) Sloth1-HA or (**B**) Sloth2-HA as prey. Blots were striped and reprobed after detection of each antigen. Arrowheads indicated expected band, asterisks indicate unknown bands. (**C**) Developmental viability assay using *tub-Gal4* to overexpress indicated transgenes throughout development. Crosses resulting in no viable adults are scored as lethal (black box).

To test whether Sloth1/2 physically interact with subunits of mitochondrial complex III, we performed co-immunoprecipitation experiments in transfected S2R+ cells. SMIM4 and C12orf73 interact with complex III subunits UQCC1 and UQCRFS1, respectively (*Zhang et al., 2020*; *Dennerlein et al., 2021*). Therefore, we tested if Sloth1 or Sloth2 could immunoprecipitate the fly homologs CG10075 (dUQCC1) or RFeSP (dUQCRFS1). Using Sloth1-FLAG as bait, we detected CG10075-HA (*Figure 8E*) and RFeSP-HA (*Figure 8F*) binding to anti-FLAG beads. In contrast, Sloth2-FLAG pulled-down CG10075-HA and RFeSP-HA weakly or was at background levels (*Figure 8E and F*). Together, these results suggest that Sloth1/2 are required for proper complex III assembly, mediated through physical interaction with complex III subunits.

## Sloth1 and Sloth2 act in a stoichiometric complex

We speculated that Sloth1 and Sloth2 could physically interact, based on the observation that both share the same loss of function phenotypes and subcellular localization. Indeed, some paralogs bind to the same protein complex (*Szklarczyk et al., 2008*) and there is a tendency for proteins in the same complex to be co-expressed (*Papp et al., 2003*). To confirm this putative interaction between Sloth1 and Sloth2, we used co-immunoprecipitation and western blotting. This revealed that Sloth1-FLAG could immunoprecipitate Sloth2-HA (*Figure 9A*), and reciprocally Sloth2-FLAG (*Figure 9B*) could immunoprecipitate Sloth1-HA. Interestingly, the levels of tagged peptide in cell lysates were higher when the opposite peptide was overexpressed (*Figure 9A and B*). Proteins in a complex commonly have important stoichiometry and unbound proteins can be degraded to preserve this balance (*Papp et al., 2003*; *Sopko et al., 2006*; *Veitia et al., 2008*; *Prelich, 2012*; *Bergendahl et al., 2019*). Furthermore, imbalanced protein complex stoichiometry can lead to haploinsufficient or dominant negative phenotypes (*Papp et al., 2003*; *Sopko et al., 2006*; *Veitia et al., 2008*; *Prelich, 2012*; *Bergendahl et al., 2019*).

To test this possibility for Sloth1/2, we overexpressed either *sloth1* or *sloth2* in vivo. Low-level ubiquitous overexpression (using *da-Gal4*) of either *UAS-sloth1* or *UAS-sloth2* cDNA had no effect on adult fly viability (*Figure 2L*). To increase expression levels, we used the strong ubiquitous driver *tub-Gal4*. Whereas *tub>sloth1* flies were viable as adults, *tub>sloth2* animals were 100% pupal lethal (*Figure 9C*). However, *tub>sloth2* animals could be rescued to adulthood by co-expression of *sloth1*. Importantly, this rescue was not due to dilution of the Gal4 transcription factor on two *UAS* transgenes, since co-expression of *UAS-tdtomato* did not rescue *tub>sloth2* lethality. Finally, *tub-Gal4* overexpression of the entire *sloth1-sloth2* bicistronic transcript resulted in viable adult flies. In all, these results suggest that Sloth1 and Sloth2 interact in a complex where their stoichiometric ratio is important for normal function.

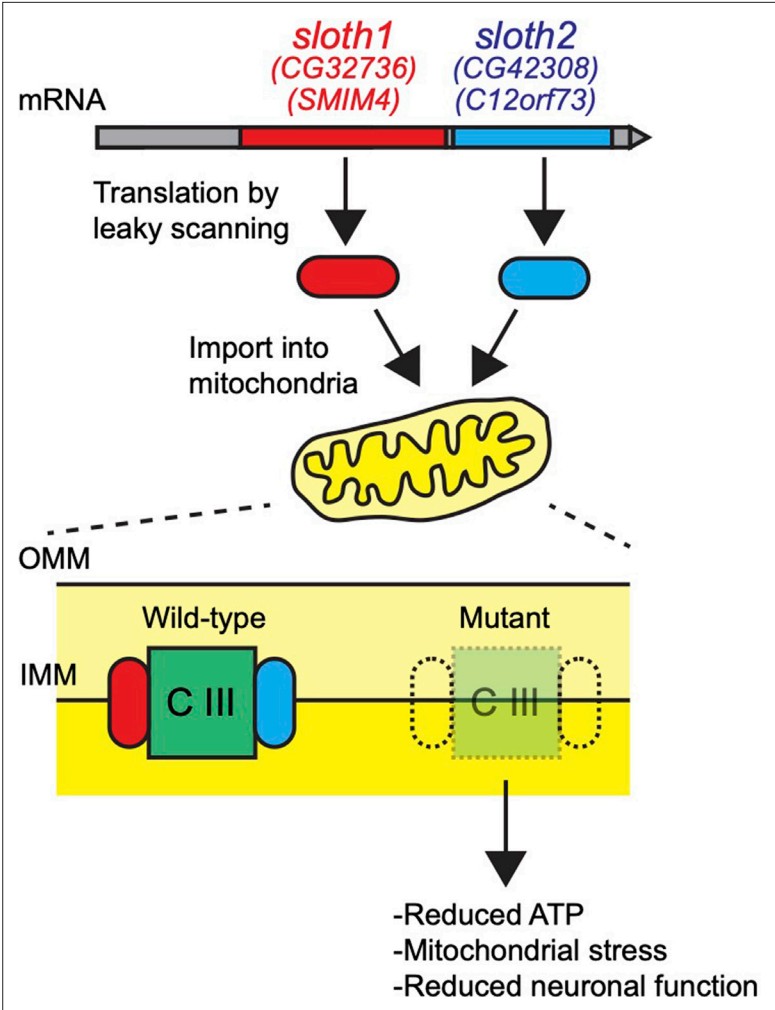

**Figure 10.** Model for Sloth1 and Sloth2 bicistronic translation and function in mitochondria.

## Discussion

Here, we have assigned new functions to two previously uncharacterized smORF peptides. Sloth1 and Sloth2 appear to be distantly related paralogs, yet each is important to support mitochondrial and neuronal function in *Drosophila*. We propose a model where Sloth1 and Sloth2 peptides are translated from the same transcript, imported into mitochondria where they interact with each other and complex III to promote its assembly (*Figure 10*). Our results are supported by three recent studies, two published during preparation of this work and one after manuscript submission, in which human Sloth1 (SMIM4) and Sloth2 (C12orf73/Brawnin) were discovered as novel mitochondrial complex III assembly factors in cultured human cells and zebrafish (*Zhang et al., 2020*; *Liang et al., 2022*; *Dennerlein et al., 2021*).

Muti-cistronic genes are relatively rare in eukaryotes, but some have been characterized in *Drosophila* (*Galindo et al., 2007*; *Magny et al., 2013*) and mammals (*Karginov et al., 2017*). Similar to operons in prokaryotes, it is thought that multicistronic transcripts allow for coordinated expression of proteins in the same pathway or complex (*Karginov et al., 2017*). Indeed, the similarity of loss-of-function phenotypes between *sloth1* and *sloth2* suggest that they function together in the same pathway/complex. Interestingly, 44/196 annotated bicistronic genes in *Drosophila* contain two ORFs with homology to each other (Flybase, DIOPT), and a recent study suggests that human bicistronic genes containing a smORF frequently encode physically interacting peptide/protein pair (*Chen et al., 2020*). Therefore, related peptides encoded on the same transcript may be a prevalent phenomenon in eukaryotes. ORF translation in multicistronic transcripts can occur by different mechanisms, such as

re-initiation of translation, IRES, or leaky ribosome scanning (*Van Der Kelen et al., 2009*). Our data and observations support leaky scanning, and we propose a model whereby both peptides are translated because *sloth1* contains a non-optimal Kozak sequence.

The presence of *sloth1* and *sloth2* orthologs in many eukaryotic species suggest that their function is likely broadly conserved. Indeed, we could rescue the lethality of *sloth1* and *sloth2* mutant flies by expressing their human counterparts. Interestingly, *Plasmodium* and *Arabidopsis* only have homologs with similarity to *sloth2*. Perhaps *sloth2* maintained functions more similar to its common ancestor with *sloth1*. We were unable to identify homologs in some eukaryotes such as yeast, although their amino acid sequence may simply be too diverged for detection using bioinformatic programs such as BLAST.

The physical interactions of Sloth1-Sloth2, Sloth1-RFeSP, and Sloth1-CG10075, and complex III assembly defects in *sloth1/2* loss of function animals, suggest that Sloth1/2 together regulate complex III assembly. Indeed, Sloth1 is bioinformatically predicted to localize to the mitochondrial inner membrane (DeepMito), and Sloth1 and Sloth2 have predicted transmembrane domains (TMHMM 2.0), suggesting they interact with complex III at the inner membrane. This is supported by data showing SMIM4 and C12orf73 are integral membrane proteins in the mitochondrial inner membrane (*Zhang et al., 2020*; *Dennerlein et al., 2021*). In addition, our data suggests that Sloth1 and Sloth2 interact in a stoichiometric manner, explaining why single mutants have the same phenotype as double mutants. This is supported by the finding that SMIM4 protein levels are dependent on the presence of C12orf73 and vice versa (*Dennerlein et al., 2021*; *Liang et al., 2022*). Perhaps maintenance of the proper ratio of Sloth1/2 is an important factor for optimal complex III assembly. Future experiments could address whether Sloth1 and Sloth2 directly bind each other, or if they require complex III subunits for physical association.

Several observations and experiments suggest that Sloth1/2 peptides do not have equivalent function. The two peptides have weak homology to each other (27% identity) and Sloth1 has 18aa (30%) more than Sloth2, suggesting divergence of function. Unlike Sloth1, Sloth2 does not have a clear mitochondrial-targeting signal. Perhaps Sloth2 has a cryptic signal that is not recognized by prediction software, or Sloth2 may be co-imported with Sloth1. Furthermore, we could not detect robust immunoprecipitation of RFeSP or CG10075 using Sloth2 as bait. Perhaps Sloth2 binds complex III indirectly through Sloth1, or Sloth 2 binds a different complex III subunit. More likely is that both Sloth1 and Sloth2 need to be present for binding to complex III, and the endogenous Sloth1 present under conditions of Sloth2-FLAG overexpression is insufficient for co-IP assays. Sloth2 may also be less stable than Sloth1, which could potentially explain why were unable to detect endogenous Sloth1 using anti-Sloth1 antibodies. Interestingly, only strong overexpression of Sloth2, and not Sloth1, was lethal to flies. Future studies may elucidate the mechanism explaining these functional differences in Sloth1/2.

Neurons have a high metabolic demand and critically depend on ATP generated from mitochondria to support processes such as neurotransmission (*Verstreken et al., 2005*; *Kann and Kovács, 2007*). Therefore, it is not unexpected that neurodegenerative diseases are frequently associated with mitochondrial dysfunction (*Golpich et al., 2017*). We find similar results in *Drosophila*, where loss of *sloth1* and *sloth2* leads to defects in mitochondrial function, impaired neuronal function, photoreceptor degeneration, and Rh1 accumulation in photoreceptors. Despite finding that the *Gal4-KI* reporter was strongly expressed in neurons and could rescue *sloth1/2* lethality, it is likely these peptides play important roles in other cell types. For example, publicly available RNA-seq data suggest that they are ubiquitously expressed (Flybase). In addition, neuronal expression of *sloth1* or *sloth2* was unable to rescue mutant lethality (*Figure 2L*). Furthermore, we observed *sloth1/2* loss of function phenotypes in dissected adult thoraxes, which are composed of mostly muscle. At present, there are no reported human disease-associated mutations in *SMIM4* and *C12orf73*. Mutations in these genes might not cause disease, or they might cause lethality. It is also possible that the lack of functional information on these genes has hampered identification of disease-associated mutations.

There is great interest in identifying the complete mitochondrial proteome (*Calvo et al., 2016*), so it is remarkable that Sloth1/2 have been largely missed in proteomic or genetic screens for mitochondrial components. For example, they are not present in bioinformatic and proteomic datasets of fly mitochondrial proteins (*Sardiello et al., 2003*; *Chen et al., 2015*), nor in genetic screens of lethal mutations on the X-chromosome affecting nervous system maintenance (*Yamamoto et al., 2014*). It is possible that the small size of these peptides lead to this discrepancy; due to less frequent mutations

in these ORFs, or fewer tryptic products for MS. It is also possible that these peptides form weak interactions with mitochondrial proteins, preventing their immunoprecipitation. Recently, human SMIM4 was identified in a proteomic screen (*Dennerlein et al., 2021*), human C12orf73 was identified in two proteomics screens (*Liu et al., 2018*; *Antonicka et al., 2020*) and a bioinformatic screen (*Zhang et al., 2020*), and mouse SMIM4 was identified in a proteomics screen (*Busch et al., 2019*).

Our discovery of *sloth1* and *sloth2* highlights the effectiveness of loss of function genetics for identifying smORF genes with important biological functions. Recent technical advances such as genome engineering (e.g. CRISPR/Cas9) and massively parallel profiling have the potential to rapidly assign functions to many uncharacterized smORFs (*Guo et al., 2018*; *Chen et al., 2020*). For example, investigation of uncharacterized smORF genes may yield additional important mitochondrial components. Indeed, there is a greater tendency for annotated human smORF peptides to localize to mitochondria (72/719, 10%) compared to the whole proteome (1228/20351, 6%) (UniProt). Interestingly, ~40 smORF peptides function at the human mitochondrial inner membrane (UniProt), such as the Complex III member UQCRQ (82aa) (*Usui et al., 1990*) and the recently described Mitoregulin/MoxI (56aa) that regulates the electron transport chain and fatty acid β-oxidation (*Makarewich et al., 2018*; *Stein et al., 2018*; *Chugunova et al., 2019*). Therefore, modulation of protein complexes in the inner mitochondrial membrane may be a common function of smORF peptides. As functional annotation of hundreds, perhaps thousands, of smORF genes is becoming easier, many new biological insights are likely to emerge from their analyses.

# Materials and methods
## Molecular cloning
Plasmid DNAs were constructed and propagated using standard protocols. Briefly, chemically competent TOP10 *E. coli.* (Invitrogen, C404010) were transformed with plasmids containing either Ampicillin or Kanamycin resistance genes and were selected on LB-Agar plates with 100 µg/ml Ampicillin or 50 µg/ml Kanamycin. Oligo sequences are in *Supplementary file 2*.

### sloth1-sloth2 expression reporters
*pMT-sloth1-RLuc* was constructed by Gibson (NEB E2611) assembly of two DNA fragments with overlapping sequence, (1) 5'UTR, *sloth1* coding sequence, and intervening sequence (*GCAAA*) were amplified from S2R+genomic DNA. (2) Plasmid backbone was amplified from *pRmHa-3-Renilla* (*Zhou et al., 2008*), which contains a *Metallothionein* promoter and coding sequence for Renilla luciferase. *pMT-sloth1-RLuc* derivatives were constructed by a PCR-based site directed mutagenesis (SDM) strategy.

### shRNA expression vector for in vivo RNAi
*pValium20-sloth1-sloth2* (aka *UAS-shRNA*, or *JAB200*) was constructed by annealing complementary oligos and ligating into *pValium20* (*Ni et al., 2011*) digested with NheI and EcoRI. See *Figure 1—figure supplement 1* for location of target site.

### sgRNA expression vectors for CRISPR/Cas9
Plasmids encoding two sgRNAs were constructed by PCR amplifying an insert and ligating into *pCFD4* (*Port et al., 2014*) digested with BbsI. sgRNAs constructed: *pCFD4-sloth1* (aka JAB203), *pCFD4-sloth2* (aka GP01169), *pCFD4-sloth1-sloth2* (aka JAB205, for dKO). See *Figure 1—figure supplement 1* for location of target sites.

### Gal4 HDR donor plasmid
*pHD-sloth1-sloth2-Gal4-SV40-loxP-dsRed-loxP* was assembled by digesting *pHD-DsRed-attP* (*Gratz et al., 2014*) with EcoRI/XhoI and Gibson assembling with four PCR amplified fragments: (1) Left homology arm from genomic DNA from *nos-Cas9[attP2]* flies. (2) *Gal4-SV40* from *pAct-FRT-stop-FRT3-FRT-FRT3-Gal4 attB* (*Bosch et al., 2015*). (3) *loxP-dsRed-loxP* from *pHD-DsRed-attP*. (4) Right homology arm from genomic DNA from *nos-Cas9[attP2]* flies.

## Custom pEntr vectors

Construction of pEntr vectors (for Gateway cloning) was performed by Gibson assembly of PCR amplified backbone from pEntr-dTOPO (Invitrogen C4040-10) and PCR amplified gene coding sequence (when appropriate, with or without stop codon). List of plasmids: *pEntr_sloth1* (from S2R+cDNA), *pEntr_sloth2* (from S2R+cDNA), *pEntr_hSMIM4* (from IDT gBlock), *pEntr_hC12orf73* (from IDT gBlock), *pEntr_sloth1-sloth2* transcript (from S2R+cDNA), *pEntr_sloth1-sloth2* genomic (from S2R+genomic DNA), and *pEntr_BFP* (from *mTagBFP2*). Derivatives of *pEntr_sloth1-sloth2* genomic that lack *sloth1* or *sloth2* coding sequence, or derivatives of *pEntr_sloth1* or *pEntr_sloth2* with or without only the N-terminal signal sequence, were generated by PCR amplifying the plasmid and reassembling the linearized plasmid (minus the desired sequence) by Gibson.

## Custom gateway expression vectors

*pMT-GW-SBP* was constructed by digesting *pMK33-SBP-C* (*Yang and Veraksa, 2017*) and *pMK33-GW* (Ram Viswanatha) with XhoI/SpeI and ligating the GW insert into digested *pMK33-SBP-C* using T4 ligase.

## Gateway cloning LR reactions

Gateway cloning reactions were performed using LR Clonase II Enzyme mix (Invitrogen 11791–020). See *Supplementary file 3* for plasmids constructed by Gateway reactions. Additional plasmids obtained were *pEntr_RFeSP* (DmCD00481962), *pEntr_CG10075* (DmCD00473802) (The FlyBi Consortium; https://flybi.hms.harvard.edu/), *pAWF* and *pAWH* (Carnegie Science/Murphy lab), *pWalium10-roe* (*Perkins et al., 2015*), and *pBID-G* (*Wang et al., 2012*).

# Fly genetics

Flies were maintained on standard fly food at 25 °C. Wild-type (WT) or control flies refers to *yw*. The *yv; attP40* strain is used as a negative control for experiments involving an shRNA or sgRNA transgene inserted into *attP40*.

Fly stocks were obtained from the Perrimon lab collection, Bloomington Stock center (indicated with BL#), or generated in this study (see below). Bloomington Stocks: *yw* (1495), *yv; P{y[+t7.7]=CaryP} attP40* (36304), *yv,P{y[+t7.7]=nos-phiC31\int.NLS}X; P{y[+t7.7]=CaryP}attP40* (25709), *P{y[+t7.7]=nos-phiC31\int.NLS}X, y(1) sc(1) v(1) sev(21); P{y[+t7.7]=CaryP}attP2* (25710), *w[1118]; Dp(1;3)DC166, PBac{y[+mDint2] w[+mC]=DC166}VK00033* (30299), *y(1) M{w[+mC]=Act5C-Cas9.P}ZH-2A w[*]* (54590), *y(1) sc[*] v(1) sev(21); P{y[+t7.7] v[+t1.8]=nos-Cas9.R}attP2* (78782), *w[*]; P{w[+mC]=UAS-2xEGFP}AH2* (6874), *w[1118]; P{w[+mC]=UAS GFP.nls}14* (4775), *y1 w*; P{tubP-GAL4}LL7/TM3, Sb1 Ser1* (5138), *MN-Gal4, UAS-mitoGFP* (42737), *MN-Gal4, UAS-nSybGFP* (9263), *UAS-tdTomato* (92759), *elav-Gal4* (8760). Perrimon Lab stocks: *w; da-Gal4, lethal/FM7-GFP*.

Transgenic flies using PhiC31 integration were made by injecting attB-containing plasmids at 200 ng/µl into integrase-expressing embryos that contained an attP landing site (attP40 or attP2). Injected adults were outcrossed to balancer chromosome lines to isolate transgenic founder flies and eventually generate balanced stocks. *pCFD4-sloth1[attP40]* (aka JAB203), *pCFD4-sloth2[attP40]* (aka GP01169), *pCFD4-sloth1-sloth2[attP40]* (aka JAB205, for dKO), *pValium20-sloth1-sloth2[attP40]* (aka *UAS-shRNA*, or *JAB200*) lines were selected with *vermillion+*. *pWalium10-sloth1[attP2], pWalium10-sloth2[attP2], pValium10-sloth2[attP40], pWalium10-hSMIM4[attP2], pWalium10-hC12orf73[attP2], pWalium10-sloth1-sloth2*transcript[attP2], *pBID-{sloth1-sloth2}[attP40], pBID-{Δsloth1-sloth2}[attP40], pBID-{sloth1-Δsloth2}[attP40]* were selected with *white+*.

*sloth1*-KO, *sloth2*-KO, and *dKO* fly lines were made by crossing sgRNA-expressing transgenic lines to *nos-Cas9[attP2]* flies, outcrossing progeny to *FM7-GFP* balancer flies, and screening progeny founder flies for deletions by PCR and Sanger sequencing.

*Gal4-KI* flies were made by injecting sgRNA plasmid (JAB205) and *pHD-sloth1-sloth2-Gal4-SV40-loxP-dsRed-loxP*, each at 200 ng/µl, into embryos expressing Cas9 in the germ line (*nos-Cas9*). Injected adults were outcrossed to *FM7-GFP* flies, progeny were screened for RFP +expression, and RFP +founder lines were confirmed by PCR for a correct knock-in.

Knockdown crosses were performed by crossing *da-Gal4* with *pValium20-sloth1-sloth2[attP40]/CyO* (aka *UAS-shRNA*, or *JAB200*) or *attP40/CyO* as a negative control. Quantification of viability was performed by counting the number of progeny with or without the CyO balancer. A Chi-square test

was used to determine if the ratio of non-balancer flies (CyO⁻) to balancer flies (CyO⁺) was significantly altered in shRNA crosses compared to control crosses. Data was analyzed using Excel and Prism.

For climbing assays, *da-Gal4/shRNA or da-Gal4/attP40* adult progeny were aged 1 week after eclosion and 10 flies were transferred into empty plastic vials without use of CO2. Climbing ability was quantified by tapping vials and recording the number of flies that climb to the top of the vial within 10 s, using video analysis. Climbing assays with the same 10 flies were performed three times and averaged. Three biological replicates were performed for each genotype. A T-Test was used to calculate statistical significance. Data was analyzed using Excel and Prism.

Somatic knockout crosses were performed by crossing *Act-Cas9* to *sgRNA[attP40]/CyO* or *attP40/CyO* as a negative control. *Act-Cas9/sgRNA[attP40]* female and male progeny were analyzed for phenotypes. Quantification of viability was performed by counting the number of progeny with or without the *CyO* balancer. A Chi-square test was used to determine if the ratio of non-balancer flies (*CyO⁻*) to balancer flies (*CyO⁺*) was significantly altered in somatic knockout crosses compared to control crosses. Male and female progeny were analyzed separately because they differ in the number of copies of the endogenous *sloth1-sloth2* loci on the X-chromosome. Data was analyzed using Excel and Prism.

Mutant and genomic rescue crosses were performed by crossing *mutant/FM7-GFP* females to genomic rescue constructs or *attP40* as a negative control. *mutant/Y* hemizygous male progeny were analyzed for phenotypes. Quantification of viability was performed by counting the number of *mutant/Y* vs *FM7GFP* male progeny. Gal4/UAS rescue crosses were performed by crossing *mutant/FM7-GFP;; da-Gal4* females to *UAS-X* lines. Additionally, *Gal4-KI/FM7-GFP* females were crossed to *UAS-X*. Rare *sloth1*-KO, *sloth2-KO, dKO,* and *Gal4-KI* hemizygous adult males normally die by sticking to the fly food after they eclose. To collect these rare mutants for further analysis (scutellar bristle images, climbing assays), we inverted progeny vials so that mutant adults fell onto the dry cotton plug once they eclose.

Overexpression crosses were performed by crossing *tub-Gal4/TM3* females to *UAS-X* lines. At least 100 *tub-Gal4/UAS-X* progeny were analyzed for phenotypes.

## Cell fractionation and mitochondrial isolation

To isolate mitochondria from S2*R*+ cells, cell pellets were resuspended in 1.1 ml hypotonic buffer (10 mM NaCl, 1.5 mM MgCl2, 10 mM Tris- HCl pH 7.5), transferred to cold glass dounce on ice, and incubated for 10 min to induce cell swelling. Cells were homogenized with 10 strokes using pestle B (tight pestle), followed by addition of 800 µl of 2.5 x homogenization buffer (525 mM mannitol, 175 mM sucrose, 12.5 mM Tris-HCl pH 7.5 and 2.5 mM EDTA). Homogenates at this step are considered whole cell lysate (WCL). WCL was centrifuged at 1,300 g for 5 min at 4 °C, supernatant transferred to a new tube, repeated centrifugation. Supernatant was transferred to a new tube and centrifuged at 17,000 g for 15 min at 4 °C. Supernatant was removed (cytoplasmic fraction) and 2 ml 1 x Homogenization buffer (210 mM mannitol, 70 mM sucrose, 5 mM Tris-HCl pH 7.5 and 1 mM EDTA) was added to the pellet. The centrifugation was repeated and 250 µl 1 x Homogenization buffer was added to the pellet (mitochondrial fraction). For SDS-PAGE comparisons of cell fractions, WCL, cytoplasmic, and mitochondria were lysed in RIPA buffer and protein concentration normalized by BCA assay (Thermo Fischer, 23227).

Mitochondrial isolation from 7-day-old adult thoraxes and whole 3rd instar larvae was modified from **Garcia et al., 2017**. Briefly, dissected adult male thoraxes or whole 3rd instar male larvae were placed into 100 µl mitochondrial isolation buffer (250 mM Sucrose, 150 mM MgCl2, 10 mM Tris-HCl pH 7.4) on ice. Thoraxes were ground using a blue pestle and a motorized pestle holder. A total of 400 µl mitochondrial isolation buffer was added to homogenized thoraxes and samples were centrifuged at 500 g at 4 °C for 5 min to pellet debris and tissues. Supernatant was transferred to a new tube and the centrifugation repeated. Supernatant was transferred to a new tube and centrifuged at 5000 g at 4 °C for 5 min to pellet mitochondria. The mitochondrial pellet was washed 2 x by adding 1 ml mitochondrial isolation buffer and repeating centrifugation at 5000 g at 4 °C for 5 min. For BN-PAGE experiments, 10 thoraxes or 10 whole 3rd instar larvae were used. For SDS-PAGE, 30 thoraxes or 30 whole 3rd instar larvae were used, and mitochondria were lysed in RIPA buffer and protein concentration normalized by BCA assay (Thermo Fischer, 23227).

## Blue native PAGE (BN-PAGE) of mitochondrial respiratory complexes

Native mitochondrial respiratory complexes were visualized by Blue Native PAGE (BN-PAGE) gels following the manufacturer's instructions protocols (Nativepage 12% Bis Tris Protein Gels, 1.0 mm, 15 well, Thermo Fisher Scientific BN1003BOX). Mitochondrial pellets from 10 thoraxes or 10 larvae were resuspended in 20 µl sample buffer cocktail (5 µl sample buffer, 8 µl 5% digitonin, 7 µl H20, 2 µl 5% Coomassie G-250 sample additive). 15 µl sample ran on each lane.

## Cell culture

*Drosophila* S2R+ cells (*Yanagawa et al., 1998*), or S2R+ cells stably expressing Cas9 and a mCherry protein trap in *Clic* (known as PT5/Cas9) (*Viswanatha et al., 2018*), were cultured at 25 °C using Schneider's media (21720–024, ThermoFisher) with 10% FBS (A3912, Sigma) and 50 U/ml penicillin strep (15070–063, ThermoFisher). S2R+ cells were transfected using Effectene (301427, Qiagen) following the manufacturer's instructions.

For generating stable cell lines MT-Sloth1-SBP, MT-Sloth2-SBP, and MT-Sloth1/2, S2R+ cells were seeded in six-well plates and transfected with *pMK33* expression plasmids (see *Supplementary file 3*). *pMK33* derived plasmids contain a Hygromycin resistance gene and a *Metallothionein* promoter to induce gene expression. After 4 days, transfected cells were selected with 200 µg/ml Hygromycin in Schneider's medium for approximately 1 month. For induction of gene expression, cells were cultured with 500 µM CuSO4 in Schneider's medium for 16 hr.

For generating KO cell lines, S2R+ Cas9 cells were transfected with *tub-GFP* plasmid (gift of Steve Cohen) and an sgRNA-expressing plasmid (*pCFD4-sloth1[attP40]* (aka JAB203), *pCFD4-sloth2[attP40]* (aka GP01169), or *pCFD4-sloth1-sloth2[attP40]* (aka JAB205, for dKO)). Forty-eight hr after transfection, cells were resuspended in fresh media, triturated to break up cell clumps, and pipetted into a cell straining FACS tube (352235 Corning). Single GFP + cells were sorted into single wells of a 96-well plate containing 50% conditioned media using an Aria-594 instrument at the Harvard Medical School Division of Immunology's Flow Cytometry Facility. Once colonies were visible by eye (3–4 weeks), they were expanded and analyzed by PCR and Sanger sequencing.

For co-immunoprecipitation experiments, S2R+ cells were transfected in 100 mm petri dishes. Four days after transfection, cells were resuspended and centrifuged at 1000 g for 10 min at 4 °C. Cell pellets were washed once with ice-cold 1 x PBS, re-centrifuged, and flash frozen in liquid nitrogen. Cell pellets were subjected to mitochondrial isolation (described above) and mitochondrial pellets were flash frozen in liquid nitrogen. Mitochondrial pellets were resuspended in 250 µl mitochondrial lysis buffer (~.5–1 ug/ul final protein concentration), incubated on ice for 30 min and centrifuged at 13,000 g for 10 min at 4 °C. The supernatant was incubated with 20 µl magnetic anti-FLAG beads (Sigma-Aldrich M8823) for 2 hr at 4 °C with gentle rocking. Beads were washed 3 x in mitochondrial lysis buffer using a magnetic stand and eluted for 30 min at 4 °C with 20 ul 3xFLAG peptide diluted at 1 mg/ml in mitochondrial lysis buffer. Mitochondrial lysis buffer: 50 mM Tris-HCl pH 7.4, 150 mM NaCl, 10% glycerol (v/v), 20 mM MgCl2, 1% digitonin (v/w) (Sigma D141), protease inhibitor (Pierce 87786), and 2 mM PMSF added immediately before use.

To measure mitochondrial respiration in S2R+ cells, we performed a Mito Stress Test on a Seahorse XFe96 Analyzer (Agilent, 103015–100). 50,000 cells were seeded into Seahorse XF96 tissue culture microplates and incubated at 25 °C overnight. One hr before analysis, cell culture media was replaced with serum-free Schneider's media and drugs were loaded into the Seahorse XFe96 Sensor Cartridge (Final concentrations: Oligomycin 1 µM, Bam15.5 µM, 1 µM Antimyzin/Rotenone 'R/A'). Seahorse analysis was performed at room temperature. Mitochondrial respiration recordings were normalized to cell number using CyQUANT (Thermo Fisher C7026) fluorescence on a plate reader. Data analysis was performed using Seahorse Wave Desktop Software 2.6, Excel, and Prism. N=6 wells for each condition. Significance was calculated using a T-Test.

To measure *MT-sloth1-RLuc* reporter expression, S2R+ cells were transfected in white opaque-bottom 96 well plates with *MT-sloth1-RLuc* (or derivatives) and *MT-FLuc* (Firefly Luciferase) (*Zhou et al., 2008*) as an internal control. Briefly, to each well, 10 ng of plasmid mix was added, then 10 µl Enhancer mix (.8 µl Enhancer + 9.2 µl EC buffer), and was incubated for 2–5 min at room temperature. 20 µl of Effectene mix (2.5 µl Effectene + 17.5 µl EC buffer) was added and incubated for 5–10 min at room temperature. 150 µl of S2R+ cells (at 3.3x10^5 cells/ml) was added gently to each well and incubated at 25 °C. After 3 days incubation, 200 µM CuSO4 was added. After 24 hr incubation, media was

gently removed from the wells by pipetting and cell luminescence was measured using the Dual-Glo assay (Promega E2920). Two luminescence normalizations were performed. First, for each sample, Renilla luminescence was normalized to Firefly luminescence (Rluc/Fluc). Next, Rluc/Fluc ratios for each sample were normalized to Rluc/Fluc ratios for wild-type *MT-sloth1-RLuc* (aka fold change Rluc/Fluc to WT). For each genotype, N=4. Significance was calculated using a T-test. Data was analyzed using Excel and Prism.

## Western blotting

Protein or cell samples were denatured in 2 x SDS Sample buffer (100 mM Tris-CL pH 6.8, 4% SDS,.2% bromophenol blue, 20% glycerol,.58 M β-mercaptoethanol) by boiling for 10 min. For western blots using glycine-based gels (*Figure 7D*, *Figure 8C–F*, *Figure 9A–B*, *Figure 6—figure supplement 2A, B, D*), denatured proteins and Pageruler Prestained Protein Ladder (Thermo Fisher Scientific 26616) were loaded into 4–20% Mini-PROTEAN TGX gels (Biorad 4561096) using running buffer (25 mM Tris, 192 mM glycine, 0.1% SDS, pH 8.3). For western blots using tricine-based gels (*Figure 6B–D*, *Figure 6—figure supplement 2C*) (to improve resolution of small peptides), denatured proteins and Precision Plus Protein Dual Xtra Prestained Protein Standards (Biorad 1610377) were loaded into 16.5% Mini-PROTEAN Tris-Tricine Gels (Biorad 4563066) using Tris/Tricine/SDS Running buffer (Biorad 1610744). Gels were ran at 100–200 V in a Mini-PROTEAN Tetra Vertical Electrophoresis Cell (Biorad 1658004). Proteins were transferred to Immobilon-FL PVDF (Millipore IPFL00010) in transfer buffer (25 mM Tris, 192 mM glycine) using a Trans-Blot Turbo Transfer System (Biorad 1704150) (Standard SD program). Resulting blots were incubated in TBST (1 x TBS + .1% Tween20) for 20 min on an orbital shaker, blocked in 5% non-fat milk in TBST for 1 hr at room temperature, and incubated with primary antibody diluted in blocking solution overnight at 4 °C. Blots were washed with TBST and incubated in secondary antibody in blocking solution for 4 hr at room temperature. Blots were washed in TBST before detection of proteins. HRP-conjugated secondary antibodies were visualized using ECL (34580, ThermoFisher). Blots were imaged on a ChemiDoc MP Imaging System (BioRad). Antibody complexes were reprobed by incubating blots with stripping buffer (Thermo Scientific 46430) following the manufacturer's instructions, re-blocked in 5% non-fat milk in TBST, and incubated with primary antibody overnight as described.

For western blots from larval brains, 3rd instar larval brains were dissected in ice cold PBS buffer with protease and phosphatase inhibitors. 10 brains per genotype were homogenized in RIPA buffer and protein concentration was measured by BCA assay (Thermo Fischer, 23227). Equal amounts of protein samples were mixed with 1 X Sample buffer (BioRad, 161–0747), boiled for 5 min, and loaded into 4–20% Mini-PROTEAN TGX gel (Bio-Rad). Gels were then transferred to nitrocellulose membranes using Bio-Rad Trans-Blot SD Semi-Dry Transfer system. Western blots using anti-Hsp60 likely recognize Hsp60A, as opposed to Hsp60B/C/D, because only Hsp60A is expressed in the larval brain (flyrnai.org/tools/dget/web).

Commercially available or published antibodies used for western blotting: rat anti-HA (1:2000, Roche 11867423001) (*Figure 9A and B*), chicken anti-HA (1:1000, ET-HA100, Aves) (*Figure 8E and F*), mouse anti-FLAG (1:1000, Sigma F1804), mouse anti-SBP (1:1000, Santa Cruz sc-101595), mouse anti-a-Tubulin (1:20,000, Sigma T5168), rabbit anti-GFP (1:5000, Invitrogen A-6455), rabbit anti-Hsp60 antibody (Abcam ab46798), mouse anti-actin (MP Biomedicals 08691002), anti-actin Rhodamine (Biorad 12004163), rabbit anti-SMIM4 (1:10,000, HPA047771), anti-UQCR-C2 (1:1000, *Murari et al., 2020*), anti-SdhA (1:1000, *Murari et al., 2020*), rabbit anti-C12orf73 (1:1000, HPA038883), anti-mouse HRP (1:3000, NXA931, Amersham), anti-rat HRP (1:3000, Jackson 112-035-062), anti-rabbit HRP (1:3000, Amersham NA934), anti-chicken HRP (1:1000, Sigma SAB3700199), anti-mouse 800 (only used in *Figure 8E and F* to detect mouse anti-FLAG) (1:5000, Invitrogen A32730). Anti-Sloth1 and Anti-Sloth2 antibodies (1:1000) were raised in rabbits (Genscript, PolyExpress Silver Package). Epitopes used: Anti-Sloth1 #1: RRLLDSWPGKKRFGC, Anti-Sloth1 #2: CEQQHLQARAANNTN, Anti-Sloth2 #1: CHSTQVDPTAKPPES, Anti-Sloth2 #2: CYKPLEDLRVYIEQE.

## Molecular biology

S2R+ cell genomic DNA was isolated using QuickExtract (QE09050, Lucigen). Fly genomic DNA was isolated by grinding a single fly in 50 µl squishing buffer (10 mM Tris-Cl pH 8.2, 1 mM EDTA, 25 mM NaCl) with 200 µg/ml Proteinase K (3115879001, Roche), incubating at 37 °C for 30 min, and 95 °C for

2 min. PCR was performed using Taq polymerase (TAKR001C, ClonTech) when running DNA fragments on a gel, and Phusion polymerase (M-0530, NEB) was used when DNA fragments were sequenced or used for molecular cloning. DNA fragments were run on a 1% agarose gel for imaging or purified on QIAquick columns (28115, Qiagen) for sequencing analysis. Sanger sequencing was performed at the DF/HCC DNA Resource Core facility and chromatograms were analyzed using Lasergene 13 software (DNASTAR).

For RT-qPCR analysis of *sloth1-sloth2* RNAi knockdown, *da-Gal4* was crossed with *attP40* or *UAS-shRNA* and ten 3rd instar larvae progeny of each genotype were flash frozen in liquid nitrogen. Frozen larvae were homogenized in 600 μl Trizol (Invitrogen 15596026) and RNA extracted using a Direct-zol RNA Miniprep kit (Zymo Research, R2050). cDNA was generated using the iScript Reverse Transcription Supermix (BioRad 1708840). cDNA was analyzed by RT-qPCR using iQ SYBR Green Supermix (BioRad 170–8880). qPCR primer sequences are listed in *Supplementary file 2*. Each qPCR reaction was performed with two biological replicates, with three technical replicates each. Data was analyzed using Bio-Rad CFX Manager, Excel, and Prism. Data from *sloth1-sloth2*-specific primers were normalized to primers that amplify *GAPDH* and *Rp49*. Statistical significance was calculated using a T-Test.

## Bioinformatic analysis

Protein similarity between fly and human Sloth1 and Sloth2 orthologs was determined using BLASTP (blast.ncbi.nlm.nih.gov) by defining the percent amino acid identity between all four comparisons. Homologs in other organisms and their gene structure were identified using a combination of BLASTP, Ensembl (https://www.ensembl.org/), HomoloGene (https://www.ncbi.nlm.nih.gov/homologene), and DIOPT (https://www.flyrnai.org/diopt). Protein accession numbers: Human *SMIM4* NP_001118239.1, Human *C12orf73* NP_001129042.1, Mouse *SMIM4* NP_001295020.1, Mouse *C12orf73 homolog* NP_001129039.1, Zebrafish *SMIM4* NP_001289975.1, Zebrafish *C12orf73 homolog* NP_001129045.1, Lamprey *SMIM4* XP_032827557.1, Lamprey *C12orf73 homolog* XP_032827559.1, *D.melanogaster CG32736* NP_727152.1, *D.melanogaster CG42308* NP_001138171.1, *Arabidopsis AT5G57080* NP_200518.1, *Arabidopsis AT4G26055* NP_001119059.1, *Plasmodium PF3D7_0709800* XP_002808771.1, Choanoflagellate (*Salpingoeca urceolata*) m.92763 (**RRichter et al., 2018**), Choanoflagellate (*Salpingoeca urceolata*) *sloth2* homolog is unannotated but present in comp15074_c0_seq2 (**Richter et al., 2018**). Sea squirt (*C. intestinalis*) *sloth1* and *sloth2* homologs are unannotated but present in *LOC100183920* XM_018812254.2. Genomic sequences for *sloth1/2* ORFs in *D. melanogaster*, Lamprey, Choanoflagellate, and Sea squirt are shown in *Supplementary file 1*.

Amino acid sequence of fly and human Sloth1/Sloth2 were analyzed for predicted domains using the following programs: MitoFates (http://mitf.cbrc.jp/MitoFates/cgi-bin/top.cgi), DeepMito (http://busca.biocomp.unibo.it/deepmito/), TMHMM 2.0 (http://www.cbs.dtu.dk/services/TMHMM/).

Amino acid sequences were aligned using Clustal Omega (https://www.ebi.ac.uk/Tools/msa/clustalo/) and visualized using Jalview (https://www.jalview.org/).

## Imaging

For imaging adult scutellar bristles, adult flies were frozen overnight and dissected to remove their legs and abdomen. Dissected adults were arranged on a white surface and a focal stack was taken using a Zeiss Axio Zoom V16. Focal stacks were merged using Helicon Focus 6.2.2.

For imaging larval brains, wandering 3rd instar larvae were dissected in PBS and carcasses were fixed in 4% paraformaldehyde for 20 min. Fixed carcasses were either mounted on slides in mounting medium (see below), or permeabilized in PBT, blocked for 1 hr in 5% normal goat serum (S-1000, Vector Labs) at room temperature, and incubated with primary antibody (anti-Elav) overnight at 4 °C, washed with PBT, incubated with secondary antibody (anti-mouse 633) for 4 hr at room temperature, washed with PBT and PBS, and incubated in mounting media (90% glycerol + 10% PBS) overnight at 4 °C. Larval brains were dissected from carcasses and mounted on a glass slide under a coverslip using vectashield (H-1000, Vector Laboratories Inc). Images of larval brains were acquired on a Zeiss Axio Zoom V16 or a Zeiss 780 confocal microscope. Images were processed using Fiji software.

For imaging the larval NMJ, wandering 3rd instar larvae were dissected as previously described (**Brent et al., 2009**). Briefly, larvae were pinned to a Sylgard-coated (Dow 4019862) petri dish, an incision was made along their dorsal surface, their cuticle was pinned down to flatten the body wall muscles, and were fixed in 4% paraformaldehyde for 20 min. Fixed carcasses were permeabilized in

PBT, blocked for 1 hr in 5% normal goat serum (S-1000, Vector Labs) at room temperature, and incubated with primary antibody overnight at 4 °C, washed with PBT, incubated with secondary antibody for 4 hr at room temperature, washed with PBT and PBS, and incubated in mounting media (90% glycerol + 10% PBS) overnight at 4 °C. Whole carcasses mounted on a glass slide under a coverslip using vectashield (H-1000, Vector Laboratories Inc). Images of the NMJ were acquired on a Zeiss Axio Zoom V16 or a Zeiss 780 confocal microscope. Images were taken from muscle 6/7 segment A2. Images were processed using Fiji software. Quantification of bouton number from NMJ stained with anti-HRP and anti-Dlg1 was performed by manual counting of boutons in an entire NMJ for wild-type (N=8) and dKO animals (N=7). A T-test was used to determine significance.

For imaging whole larvae, wandering 3rd instar larvae were washed with PBS and heat-killed for 5 min on a hot slide warmer to stop movement. Larvae were imaged using a Zeiss Axio Zoom V16 fluorescence microscope.

For imaging the adult brain, ~1-week-old adult flies were dissected in PBS and whole brains were fixed in 4% paraformaldehyde for 20 min. Fixed brains were permeabilized in PBT, blocked for 1 hr in 5% normal goat serum (S-1000, Vector Labs) at room temperature, incubated with anti-HRP 647 overnight at 4 °C, washed with PBT and PBS, and incubated in mounting media (90% glycerol + 10% PBS) overnight at 4 °C. Adult brains were mounted on glass slides under a coverslip using vectashield (H-1000, Vector Laboratories Inc). Images of adult brains were acquired on a Zeiss 780 confocal microscope. Images were processed using Fiji software.

For confocal microscopy of adult photoreceptors, the proboscis was removed and the head was pre-fixed with 4% formaldehyde in PBS for 30 min. After pre-fixation, eyes were removed from the head and fixed an additional 15 min. Fixed eyes were washed with PBS 3 x for 10 min each and permeabilized in 0.3% Triton X-100 in PBS for 15 min. Permeabilized, fixed samples were blocked in 1 X PBS containing 5% normal goat serum (NGS) and 0.1% Triton X-100 for 1 hr (PBT). Samples were incubated in primary antibody diluted in PBT overnight at 4 °C, washed 3 x with PBT, and incubated in secondary antibodies in NGS for 1 hr at room temp the next day. Following secondary antibody incubation, samples were washed with PBS and were mounted on microscope slides using vectashield. Samples were imaged with LSM710 confocal with 63 X objective and processed using Fiji software.

S2R+ cells transfected with Sloth1-FLAG or Sloth2-FLAG were plated into wells of a glass-bottom 384 well plate (6007558, PerkinElmer) and allowed to adhere for 2 hr. Cells were fixed by incubating with 4% paraformaldehyde for 30 min, washed with PBS with.1% TritonX-100 (PBT) 3x5 min each, blocked in 5% Normal Goat Serum (NGS) (S-1000, Vector Laboratories) in PBT for 1 hr at room temperature, and incubated in primary antibodies diluted in PBT-NGS overnight at 4 °C on a rocker. Wells were washed in PBT, incubated with secondary antibodies and DAPI and washed in PBS. Plates were imaged on an IN Cell Analyzer 6000 (GE) using a 20 x or 60 x objective. Images were processed using Fiji software.

List of antibodies and chemicals used for tissue staining: rat anti-Elav (1:50, DSHB, 7E8A10), goat anti-HRP 647 (1:400, Jackson Immunoresearch, 123-605-021), mouse anti-ATP5α (1:500, Abcam, ab14748), DAPI (1:1000, Thermo Fisher, D1306), rabbit anti-FLAG (1:1000, Sigma, F7425), mouse anti-FasII (1:25, DSHB, 1D4), mouse anti-brp (1:25, DSHB, nc82), mouse anti-Dlg1 (1:250, DSHB, 4F3), anti-mouse 633 (1:500, A-21052, Molecular Probes), mouse monoclonal anti-Rh1 (1:50, DSHB 4C5), Phalloidin conjugated with Alexa 488 (1:250, Invitrogen A12379).

## Transmission electron microscopy (TEM) of adult photoreceptors

TEM of *Drosophila* adult retinae were performed following standard electron microscopy procedures using a Ted Pella Bio Wave processing microwave with vacuum attachments. Briefly, whole heads were dissected in accordance to preserve the brain tissue. The tissue was covered in 2% paraformaldehyde, 2.5% Glutaraldehyde, in 0.1 M Sodium Cacodylate buffer at pH 7.2. After dissection, the heads were incubated for 48 hr in the fixative on a rotator at 4 °C. The pre-fixed heads were washed with 3 X millipore water followed by secondary fixation with 1% aqueous osmium tetroxide, and rinsed again 3 X with millipore water. To dehydrate the samples, concentrations from 25%–100% of Ethanol were used, followed by Propylene Oxide (PO) incubation. Dehydrated samples are infiltrated with gradual resin:PO concentrations followed by overnight infiltration with pure resin. The samples were embedded into flat silicone molds and cured in the oven at 62 °C for 3–5 days, depending on the atmospheric humidity. The polymerized samples were thin-sectioned at 48–50 nm and stained with 1% uranyl acetate for 14 min followed

by 2.5% lead citrate for two minutes before TEM examination. Retina were viewed in a JEOL JEM 1010 transmission electron microscope at 80kV. Images were captured using an AMT XR-16 mid-mount 16 mega-pixel digital camera in Sigma mode. Three animals per genotype per condition were used for TEM. At least 30 photoreceptors were used for organelle quantifications. Quantification of photoreceptor number, number of aberrant photoreceptors, and number of mitochondria per photoreceptor, was performed in Prism. Significance was calculated using a T-Test.

## Electrical recordings

### Intracellular recording from larval NMJ

3rd instar larval NMJ recordings were performed as described previously (*Ugur et al., 2017*). Briefly, free moving larvae are dissected in HL3.1 buffer without $Ca^{2+}$. Recordings were performed by stimulating the segmental nerve innervating a hemisegment A3, Muscle 6/7 through a glass capillary electrode filled with HL3.1 with 0.75 mM $Ca^{2+}$. There were no differences in input resistance, time constant $\tau$, and resting membrane potential among different genotypes tested. Repetitive stimulations were performed at 10 Hz and were reported relative to the first excitatory junction potential (EJP). Data were processed with Mini Analysis Program by Synaptosoft, Clampfit, and Excel. At least five animals were used per each genotype per essay. Significance was calculated using a T-Test.

### Electroretinograms (ERGs)

ERGs were recorded according to *Jaiswal et al., 2015*. Briefly, flies were immobilized on a glass slide with glue. Glass recording electrodes, filled with 100 mM NaCl, were placed on the surface of the eye to record field potential. Another electrode placed on the humerals served as a grounding electrode. Before recording ERGs, flies were adjusted to darkness for three minutes. Their response to light was measured in 1 s. intervals for 30 s. To test if the flies can recover from repetitive stimulation, we recorded ERGs after 30 s. and 1 min constant darkness following repetitive stimulation. Data were processed with AXON-pCLAMP8.1. At least 6 animals were used per each genotype per essay. Significance was calculated using a T-Test.

## Measurement of ATP levels from larvae

Ten 3rd$^d$ instar larvae were snap frozen with liquid nitrogen in a 1.5 mL centrifuge tube. Following freezing, samples were homogenized in 100 µl of 6 M guanidine-HCl in extraction buffer (100 mM Tris and 4 mM EDTA, pH 7.8) to inhibit ATPases, and boiled for 3 min. The samples were centrifuged to remove cuticle. Supernatant was serially diluted with extraction buffer and protein concentration was measured using a BCA kit (Thermo Fischer, 23227). For each genotype, ATP levels were measured from equal protein amounts using an Invitrogen ATP detection kit (Invitrogen, A22066) according to their protocol. N=3 experiments, biological triplicates per genotype per experiment. Significance was calculated using a T-Test.

## Acknowledgements

We thank the TRiP and DRSC for help generating transgenic flies, Dr. Marcia Haigis for use of a Seahorse XF analyzer, Claire Hu, Tera Levin, and Dan Richter for bioinformatics help, Lucy Liu for assistance mounting larvae to image the NMJ, Thai LaGraff for help with qPCR, and Rich Binari and Cathryn Murphy for general assistance. We thank members of the BDGP for discussions. We also thank the HMS MicRoN (Microscopy Resources on the North Quad) Core. JAB was supported by the Damon Runyon Foundation. This work was supported by NIH grants R01GM084947, R01GM067761, R24OD019847, and NHGRI HG009352 (SEC). NP is an investigator of the Howard Hughes Medical Institute.

## Additional information

### Competing interests

Hugo J Bellen: Reviewing editor, eLife. The other authors declare that no competing interests exist.

## Funding

| Funder | Grant reference number | Author |
| --- | --- | --- |
| Damon Runyon Foundation | DRG:2258-16 | Justin A Bosch |
| National Institutes of Health | R01GM084947 | Norbert Perrimon |
| National Institutes of Health | R01GM067761 | Norbert Perrimon |
| National Institutes of Health | R24OD019847 | Norbert Perrimon |
| National Institutes of Health | NHGRI HG009352 | Susan Celniker |
| National Institutes of Health | T32GM007748 | Justin A Bosch |

The funders had no role in study design, data collection and interpretation, or the decision to submit the work for publication.

## Author contributions

Justin A Bosch, Conceptualization, Resources, Data curation, Software, Formal analysis, Supervision, Funding acquisition, Validation, Investigation, Visualization, Methodology, Writing – original draft, Writing – review and editing; Berrak Ugur, Zhongyuan Zuo, Investigation, Visualization; Israel Pichardo-Casas, Visualization, Methodology; Jordan Rabasco, Felipe Escobedo, Investigation; Ben Brown, Susan Celniker, Conceptualization, Supervision, Funding acquisition; David A Sinclair, Hugo J Bellen, Supervision; Norbert Perrimon, Supervision, Funding acquisition, Project administration, Writing – review and editing

## Author ORCIDs

Justin A Bosch ![ORCID] http://orcid.org/0000-0001-8499-1566
Berrak Ugur ![ORCID] http://orcid.org/0000-0003-4806-8891
Felipe Escobedo ![ORCID] http://orcid.org/0000-0002-6830-9210
Hugo J Bellen ![ORCID] http://orcid.org/0000-0001-5992-5989
Norbert Perrimon ![ORCID] http://orcid.org/0000-0001-7542-472X

## Decision letter and Author response

Decision letter https://doi.org/10.7554/eLife.82709.sa1
Author response https://doi.org/10.7554/eLife.82709.sa2

# Additional files

## Supplementary files

• Supplementary file 1. Genomic sequence of *sloth1-sloth2* homologs in *D. melanogaster*, *S. urceolata*, *P. marinus*, and *C. intestinalis*.
• Supplementary file 2. Oligo and dsDNA sequences.
• Supplementary file 3. Gateway cloning plasmid list.
• Supplementary file 4. Raw gel and western images.
• Transparent reporting form

## Data availability

The current manuscript did not generate any datasets. Raw gel and western image source files are present in Supplementary File 4, which can be downloaded at: https://datadryad.org/stash/share/RIjsb5EcXwJx6egHdr5p61MQ0q7aRScHQKSPpevuP4Q.

The following dataset was generated:

| Author(s) | Year | Dataset title | Dataset URL | Database and Identifier |
|---|---|---|---|---|
| Perrimon N | 2022 | Two neuronal peptides encoded from a single transcript regulate mitochondrial complex III in Drosophila | https://dx.doi.org/10.5061/dryad.83bk3j9vc | Dryad Digital Repository, 10.5061/dryad.83bk3j9vc |

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
