## [Editor Report]

The paper identifies two small protein products, Sloth1 and Sloth2, that form a complex and perform an important role in respiratory metabolism. These important findings contribute to the functional understanding of small proteins as well as to broadening the knowledge of the physiology of mitochondria, especially in the tissues of high energy demand and unique metabolic profiles such as neurons. Convincing and rich evidence is provided that Sloth1/2 is involved in maintaining the functionality of the respiratory chain, specifically of Complex III.

---

## [Decision Letter]

**Decision letter after peer review:**

[Editors’ note: the authors submitted for reconsideration following the decision after peer review. What follows is the decision letter after the first round of review.]

Thank you for submitting your work entitled "Two neuronal peptides encoded from a single transcript regulate mitochondrial function in *Drosophila*" for consideration by *eLife*. Your article has been reviewed by 2 peer reviewers, one of whom is a member of our Board of Reviewing Editors, and the evaluation has been overseen by a Reviewing Editor and a Senior Editor. The reviewers have opted to remain anonymous.

Our decision has been reached after consultation between the reviewers. Based on these discussions and the individual reviews below, we regret to inform you that your work will not be considered further for publication in *eLife*.

The reviewers agree that the findings presented in your study are potentially very interesting and they appreciate the organismal analyses presented in the manuscript that lead to a clear conclusion on the importance of Sloth1/Sloth2 peptides for neuronal function in *D. melanogaster*.

However, as you will see from the comments appended below they both point to more substantive opportunity for deeper analyses that would result in a better molecular understanding of Sloth1/Sloth2 function, localization and partners. Should you be able to do this and still want to consider *eLife*, we will be happy to consider a fresh submission.

*Reviewer #1:*

In this work, the authors discovered two small proteins that are produced from one mRNA and bear sequence similarity to each other. These proteins localize to mitochondria forming a complex, with an important role in respiration and/or ATP production. The study uses *Drosophila melanogaster* and based on the sequence and complementation analyses, the peptides are likely functionally conserved. These data contribute to functional understanding of ORFs encoding small peptides or proteins as well as to broadening the knowledge on components involved in physiology of mitochondria, specific for the tissues of extreme energy demand and unique metabolic profile such as neurons.

The authors provide a nice and complete analysis of animals lacking Sloth1 and Sloth2 on organismal and physiological level. The results demonstrate that these two peptides are essential for mitochondrial function in neurons. Their absence or depletion leads to a significant decrease in performance, degeneration and death. Furthermore, they seem to perform similar function in other organisms, such as human.

Yet, there are several major concerns that lead to the conclusion that the manuscript could further explore function at the molecular level. Despite very interesting results that clearly identify important players for mitochondrial function in neurons, there is is little or no molecular and biochemical analyses of the described phenotypes. For example, there is no analysis included on how the particular respiratory complexes appear, or more broadly what is wrong with mitochondria in the animals lacking the Sloth complex. For example, the authors could have more conclusively addressed the localization of the Sloth complex as well as its interaction partners. In summary, despite very promissing data, the current work falls short of presenting insights to the molecular role the Sloth complex in mitochondria.

*Reviewer #2:*

Bosch et al. describe the identification of two smORFs in *Drosophila*, Sloth1 and Sloth2, that are encoded on the same transcript. Deletion of the proteins results in animal lethality. Localization studies using tagged Sloth1 and 2 imply mitochondrial localization and analysis of mutant flies indicate a defect in the activity of the respiratory chain resulting in a decreased ATP level.

While the analyses of smORFs is of general interest and will likely reveal several new proteins with important functions, the manuscript by Bosch et al. does not provide sufficient evidence to support mitochondrial localization and regulation of mitochondrial functions by Sloth1 and Sloth2. The analyses are mostly descriptive, a very useful start, but do not provide insight into the proteins' functions. The endogenous proteins' localization and functions would be a valuable addition.

1. The title states that Sloth1 and Sloth2 regulate mitochondrial function, however, the provided data only show that mitochondrial respiration is compromised upon loss of the proteins and that ATP levels are decreased. This is not sufficient experimental support to state that the two proteins regulate mitochondrial functions. Unfortunately, many other statements in the text are also overreaching and not supported by experimental data, e.g. the discussion starts with the statement "Here, we have assigned new functions…", but functional analyses of Sloth1 and Sloth2 are not provided.

2. Similarly, the "impact statement" declares that Sloth1 and Sloth2 "are ancient paralogues". However, this is suggested mainly based on sequence similarities (see e.g. the abstract) and should therefore not be claimed as a main finding in the initial impact statement.

3. The authors use prediction programs to obtain information about the cellular localization of Sloth1 and Sloth2. However, several points in this analysis raise concerns: In figure 6A the results of the predictions programs are stated to be "Yes" or "No". However, prediction programs compute probabilities, e.g. MitoFates and MitoProtII give a "probability of presequence" or "probability for export to mitochondria". The authors must therefore have used cut-offs to assign which probability is a yes and no, but these thresholds are not reported in the manuscript. This is problematic as e.g. the MitoProtII prediction score for Sloth1 is only 0.4384, which is low and does not support mitochondrial localization, but the authors nevertheless assigned a "yes" to this program.

Furthermore, prediction programs can give a hint of a protein's localization, but experimental data to validate these predictions are required. The most reliable program is MitoFates as it was trained on authentic mitochondrial protein N-termini identified in N-proteomic MS analyses. The other programs are often trained on only a handful of model substrates or have other limitations e.g. iPSORT only analyses the first 30 N-terminal amino acids. Given the weak predictions of a mitochondrial localization of Sloth1 and Sloth2, experimental data is required. The experimental approach in the manuscript is microscopy of FLAG-tagged and overexpressed Sloth1 and Sloth2 that co-localizes with ATP5alpha. However, both – overexpression and tagging – often results in protein mislocalization, especially of mitochondrial proteins. Therefore, support for mitochondrial localization of Sloth1 and Sloth2 is missing. Actually, the most convincing data on a possible mitochondrial localization of Sloth2 is the recent publication on its human homologue BRAWNIN (Zhang et al. Nat commun 2020), in which the authentic non-tagged protein is localized to the IM. In contrast Bosch et al. do not provide any data using the non-tagged protein expressed from its endogenous promoter.

4. Based on CoIP experiments and MS analysis Tim8 and Tim13 are identified as interacting proteins. Unfortunately, the MS data is not provided in the manuscript, which is very unusual. How many other proteins, e.g. from other compartments were identified? It is stated that Tim8 and Tim13 were the only mitochondrial proteins identified in the MS data. This is somehow surprising as the inner mitochondrial membrane is a crowded compartment. e.g. the study by Zhang et al. that characterized the human Sloth2 homologue BRAWNIN (Zhang et al. Nat. commun. 2020) detected an interaction with respiratory chain complex III. The MS data are then further validated by CoIP and immunodecoration. Unfortunately, the analyses are not conclusive. Both interacting partners are tagged and therefore likely overexpressed. Input and elution fractions are not shown in the same panel, which makes assessment of efficiency not possible and control proteins are missing in the elution. The localization of Sloth1 and Sloth2 to mitochondria and the interaction with mitochondrial proteins is therefore not convincing and requires data from un-tagged proteins, exact localization studies (as performed in Zhang et al., a prediction by DeepMito is not sufficient to deduce IM localization) and well-controlled interactome studies.

5. Secretion of Sloth2 should have been controlled with brefeldin A, which inhibits bona fide secretion (overexpression of proteins is known to often result in secretion).

6. The authors imply that a stress response could be triggered and show increased protein levels of the matrix chaperone Hsp60. A mitochondrial loading controls is missing here (Figure 7d), especially as numbers of mitochondria are changing according to microscopy analysis.

7. At several passages in the text a finding is described with the addition (not shown). I think that all data should be shown, e.g. in the supplemental figures.

---

## [Author Response]

[Editors’ note: the authors resubmitted a revised version of the paper for consideration. What follows is the authors’ response to the first round of review.]

Reviewer #1:In this work, the authors discovered two small proteins that are produced from one mRNA and bear sequence similarity to each other. These proteins localize to mitochondria forming a complex, with an important role in respiration and/or ATP production. The study uses *Drosophila melanogaster* and based on the sequence and complementation analyses, the peptides are likely functionally conserved. These data contribute to functional understanding of ORFs encoding small peptides or proteins as well as to broadening the knowledge on components involved in physiology of mitochondria, specific for the tissues of extreme energy demand and unique metabolic profile such as neurons.The authors provide a nice and complete analysis of animals lacking Sloth1 and Sloth2 on organismal and physiological level. The results demonstrate that these two peptides are essential for mitochondrial function in neurons. Their absence or depletion leads to a significant decrease in performance, degeneration and death. Furthermore, they seem to perform similar function in other organisms, such as human.Yet, there are several major concerns that lead to the conclusion that the manuscript could further explore function at the molecular level. Despite very interesting results that clearly identify important players for mitochondrial function in neurons, there is is little or no molecular and biochemical analyses of the described phenotypes. For example, there is no analysis included on how the particular respiratory complexes appear, or more broadly what is wrong with mitochondria in the animals lacking the Sloth complex. For example, the authors could have more conclusively addressed the localization of the Sloth complex as well as its interaction partners. In summary, despite very promissing data, the current work falls short of presenting insights to the molecular role the Sloth complex in mitochondria.

The authors want to thank Reviewer #1 for the compliments and helpful comments. We included new data in the manuscript that specifically addresses the concerns.

1. To gain more evidence supporting Sloth1/2 localization to mitochondria, we performed cell fractionation assays in cultured S2R+ cells (Figure 6 and Sup. Figure 8), in which whole cell lysates (WCL) were separated into cytoplasmic and mitochondrial fractions. We showed that transfected tagged Sloth1 and Sloth2 (Sloth1-FLAG, Sloth2-FLAG, Sloth1-SBP, Sloth2-SBP) are enriched in the mitochondria fractions. Similarly, transfected untagged Sloth1 and Sloth2 are enriched in mitochondrial fractions, detected using anti-Sloth1 and anti-Sloth2 antibodies that are described in this revision. Finally, in mitochondrial fractions isolated from whole 3^rd^ instar larvae or adult thorax muscle, we show that endogenous Sloth1 is detected as 10-15kDa band, based on its absence in sloth1/2 loss of function samples.

2. To test if *sloth1/2* loss function animals have defects in respiratory complex function, we visualized native respiratory complexes using BN-PAGE, or denatured respiratory complex subunits by SDS-PAGE, in adult thorax muscle and whole 3^rd^ instar larvae (Figure 8). In both experiments, complex III assembly is disrupted by Sloth1/2 loss of function.

3. To address the issue of Sloth1/2 interaction partners, we performed new co-IP experiments (Figure 8). Sloth1 physically interacts with two complex III subunits, CG10075 (fly UQCC1) and RFeSP (fly UQCRFS1). In addition, we showed in our first submission that Sloth1 physically interacts with Sloth2. Together, these data suggest Sloth1 and Sloth2 bind to complex III, with Sloth2 perhaps binding indirectly via Sloth1.

Reviewer #2:Bosch et al. describe the identification of two smORFs in *Drosophila*, Sloth1 and Sloth2, that are encoded on the same transcript. Deletion of the proteins results in animal lethality. Localization studies using tagged Sloth1 and 2 imply mitochondrial localization and analysis of mutant flies indicate a defect in the activity of the respiratory chain resulting in a decreased ATP level.While the analyses of smORFs is of general interest and will likely reveal several new proteins with important functions, the manuscript by Bosch et al. does not provide sufficient evidence to support mitochondrial localization and regulation of mitochondrial functions by Sloth1 and Sloth2. The analyses are mostly descriptive, a very useful start, but do not provide insight into the proteins' functions. The endogenous proteins' localization and functions would be a valuable addition.

We thank Reviewer #2 for the helpful comments. In our revision, we added new experiments that increase support for our model that Sloth1 and Sloth2 are imported to mitochondria and regulate mitochondrial function (see response to Reviewer #1). See also responses below to individual points.

1. The title states that Sloth1 and Sloth2 regulate mitochondrial function, however, the provided data only show that mitochondrial respiration is compromised upon loss of the proteins and that ATP levels are decreased. This is not sufficient experimental support to state that the two proteins regulate mitochondrial functions. Unfortunately, many other statements in the text are also overreaching and not supported by experimental data, e.g. the discussion starts with the statement "Here, we have assigned new functions…", but functional analyses of Sloth1 and Sloth2 are not provided.

We toned down such claims in the text, such as removed the statement “Here, we have assigned new functions…” from the discussion and the word “function” from the title. Conversely, we provided new functional analysis in the revision.

2. Similarly, the "impact statement" declares that Sloth1 and Sloth2 "are ancient paralogues". However, this is suggested mainly based on sequence similarities (see e.g. the abstract) and should therefore not be claimed as a main finding in the initial impact statement.

Changed “ancient” to “presumed” in the Impact Statement.

3. The authors use prediction programs to obtain information about the cellular localization of Sloth1 and Sloth2. However, several points in this analysis raise concerns: In figure 6A the results of the predictions programs are stated to be "Yes" or "No". However, prediction programs compute probabilities, e.g. MitoFates and MitoProtII give a "probability of presequence" or "probability for export to mitochondria". The authors must therefore have used cut-offs to assign which probability is a yes and no, but these thresholds are not reported in the manuscript. This is problematic as e.g. the MitoProtII prediction score for Sloth1 is only 0.4384, which is low and does not support mitochondrial localization, but the authors nevertheless assigned a "yes" to this program.Furthermore, prediction programs can give a hint of a protein's localization, but experimental data to validate these predictions are required. The most reliable program is MitoFates as it was trained on authentic mitochondrial protein N-termini identified in N-proteomic MS analyses. The other programs are often trained on only a handful of model substrates or have other limitations e.g. iPSORT only analyses the first 30 N-terminal amino acids. Given the weak predictions of a mitochondrial localization of Sloth1 and Sloth2, experimental data is required. The experimental approach in the manuscript is microscopy of FLAG-tagged and overexpressed Sloth1 and Sloth2 that co-localizes with ATP5alpha. However, both – overexpression and tagging – often results in protein mislocalization, especially of mitochondrial proteins. Therefore, support for mitochondrial localization of Sloth1 and Sloth2 is missing. Actually, the most convincing data on a possible mitochondrial localization of Sloth2 is the recent publication on its human homologue BRAWNIN (Zhang et al. Nat commun 2020), in which the authentic non-tagged protein is localized to the IM. In contrast Bosch et al. do not provide any data using the non-tagged protein expressed from its endogenous promoter.

The authors appreciate the Reviewer’s careful considerations about mitochondrial targeting sequence prediction. We apologize for not reporting score cut-offs, as well as the mistake with the MitoProtII Sloth1 prediction score. To address these issues, we (1) deleted Figure panel 6A and instead report the prediction results in the main text. (2) Only give the results using MitoFates and DeepMito (no mitochondrial targeting results using TargetP, DeepLoc, PSORT, Busca, iPSORT, MitoProtII) (3) report the program scores. Now, for Sloth1, Sloth2, SMIM4, and C12orf73, only SMIM4 scores as mitochondrial using MitoFates (score.842) and only Sloth1 and SMIM4 score as mitochondrial inner membrane using DeepMito (.93 and.73). See revised text from Results section:

“SMIM4 has a predicted mitochondrial targeting sequence using MitoFates (FUKASAWA et al. 2015) (0.842), but C12orf73, Sloth1, and Sloth2 do not (.0016, 0.016, 0.009, respectively). In addition, SMIM4 and Sloth1 are predicted to localize to the mitochondrial inner membrane using DeepMito (0.93 and 0.73, respectively), but C12orf73 and Sloth2 are not (0.66 and 0.49, respectively) (SAVOJARDO et al. 2020).”

To address Reviewer #2’s concern about limited experimental evidence for Sloth1 and Sloth2 mitochondrial localization, we performed additional experiments to show that overexpressed tagged and untagged forms of Sloth1 and Sloth2 in cultured S2R+ cells are enriched in mitochondrial fractions (Figure 6 and Supplemental Figure 8) (also see above response to Reviewer #1). While we were also able to detect endogenous Sloth1 in mitochondrial fractions from larvae and adult thoraxes by western blot (also see above response to Reviewer #1), we failed to obtain similar results for endogenous Sloth2. In addition, antibody staining of neuronal tissue using anti-Sloth1 and anti-Sloth2 did not yield positive results (Supplemental Figure 7). The anti-Sloth1 and anti-Sloth2 antibodies were raised in rabbits, where two different antigenic peptides for Sloth1 were used to immunize two different rabbits, and two different antigenic peptides for Sloth2 were used to immunize two different rabbits (a total of four antibodies tested). Our negative results may be due to the antibodies being poor quality or perhaps the Sloth1 and Sloth2 peptides are inherently difficult to detect.

4. Based on CoIP experiments and MS analysis Tim8 and Tim13 are identified as interacting proteins. Unfortunately, the MS data is not provided in the manuscript, which is very unusual. How many other proteins, e.g. from other compartments were identified? It is stated that Tim8 and Tim13 were the only mitochondrial proteins identified in the MS data. This is somehow surprising as the inner mitochondrial membrane is a crowded compartment. e.g. the study by Zhang et al. that characterized the human Sloth2 homologue BRAWNIN (Zhang et al. Nat. commun. 2020) detected an interaction with respiratory chain complex III. The MS data are then further validated by CoIP and immunodecoration. Unfortunately, the analyses are not conclusive. Both interacting partners are tagged and therefore likely overexpressed. Input and elution fractions are not shown in the same panel, which makes assessment of efficiency not possible and control proteins are missing in the elution. The localization of Sloth1 and Sloth2 to mitochondria and the interaction with mitochondrial proteins is therefore not convincing and requires data from un-tagged proteins, exact localization studies (as performed in Zhang et al., a prediction by DeepMito is not sufficient to deduce IM localization) and well-controlled interactome studies.

We apologize for not providing our MS data in the first submission, this was an oversight. In the current revision, we have removed experiments involving MS data completely for the following reasons, (1) the large-scale pulldowns were performed on whole cell lysates as opposed to isolated mitochondria, perhaps increasing background and causing false negatives. The large-scale pulldowns were performed on whole cell lysates because this experiment was conducted before we suspected the peptides localized to mitochondria. (2) Two recent studies in human cell culture reported that SMIM4 and C12orf73 interact with complex III in mitochondria. Therefore, we directly tested for physical interaction of Sloth1 and Sloth2 with fly complex III subunits (Figure 8), which revealed that Sloth1 interacts with CG10075 (fly UQCC1) and RFeSP (fly UQCRFS1) (see also above response to Reviewer #1).

5. Secretion of Sloth2 should have been controlled with brefeldin A, which inhibits bona fide secretion (overexpression of proteins is known to often result in secretion).

We removed our secretion data from the revised manuscript because (1) the data does not contribute to this story about their role in the mitochondria, (2) the possible secretion of Sloth2 may be explored in a future study, and (3) SignalP 5.0 was used in the original manuscript to predict that Sloth2 had a secretion signal (score 0.5357), whereas a newer version of SignalP (6.0) does not predict Sloth2 to be a secreted protein (score.396).

6. The authors imply that a stress response could be triggered and show increased protein levels of the matrix chaperone Hsp60. A mitochondrial loading controls is missing here (Figure 7d), especially as numbers of mitochondria are changing according to microscopy analysis.

We appreciate this comment and agree that the mitochondrial loading control on western blots from 3^rd^ instar larvae brains is preferable. However, we did not include the control in our revised manuscript since we saw no obvious change in mitochondria number when staining using anti-ATP5alpha in the larval brain (Supplemental Figure 7) or in the neuromuscular junction of larval motor neurons (Supplemental Figure 4).

7. At several passages in the text a finding is described with the addition (not shown). I think that all data should be shown, e.g. in the supplemental figures.

We removed all instances of “(not shown)” from the text, outlined below.

“In addition, a full-length transcript containing both smORFs is present in the cDNA clone RE60462 (GenBank Acc# AY113525), which was derived from an embryonic library (STAPLETON et al. 2002), and we detected the fulllength bicistronic transcript from total RNA 3rd instar larvae by RT-PCR amplification (not shown).”

We supplied the data in the new Supplemental Figure 1.

“Furthermore, most escaper adults had short scutellar bristles (Figure 2H) and frequently appeared sluggish (not shown).”

The authors consider this as useful qualitative information and the “(not shown)” was simply deleted. The data is consistent with quantified data using RNAi knockdown and KO lines.

“First, transheterozygous female flies (sloth1-KO/+, sloth2-KO/+) were viable and had normal scutellar bristles (not shown).”

The authors consider this as useful qualitative information and the “(not shown)” was simply deleted.

“Gal4-KI mobility defects and lethality could be rescued by expressing the entire bicistronic transcript (UAS-sloth1sloth2) (Figure 2J, L), or coexpression of both smORFs as cDNA (UAS-sloth1 and UAS-sloth2) (Figure 2L, not shown).”

The “(not shown)” was a mistaken reference and was simply deleted.

“Therefore, we directly visualized their subcellular location. We raised antibodies to Sloth1 and Sloth2, but were unable to detect the endogenous peptides by immunostaining and western blotting (not shown).”

We report on our data with anti-Sloth1 and anti-Sloth2 antibodies in the revised manuscript. This entire sentence is no longer present in the revised manuscript.

“Mutations in *Drosophila* mitochondrial genes are known to cause phenotypes that are reminiscent of loss of sloth1 and sloth2, such as pupal lethality, developmental delay (not shown), reduced neuronal activity, photoreceptor degeneration, and Rh1 accumulation in photoreceptors (JAISWAL et al. 2015).”

Loss of sloth1 or sloth2 causes a developmental delay (e.g. sloth1/2 KO flies), but we did not quantify this in either the original or revised manuscript. Therefore, we deleted the text “, developmental delay (not shown)” in the revised manuscript.

“Finally, we tested for sloth1 or sloth2 overexpression phenotypes in vivo. Low-level ubiquitous overexpression (using da-Gal4) of either sloth1 or sloth2 had no effect on fly viability or bristle length (Figure 2L, not shown). To increase expression levels, we used the strong ubiquitous driver tub-Gal4. Whereas tub>sloth1 flies were viable (Figure 8G) and had normal bristle length (not shown), tub>sloth2 flies were 100% pupal lethal (Figure 8G). However, raising tub>sloth2 flies at 18°C, which decreases Gal4/UAS expression, produced escaper adults that had short scutellar bristles (not shown), reminiscent of loss of function of either sloth1 or sloth2 (Figure 2K). Hence, an imbalance in complex stoichiometry caused by overexpression of one member of the complex disrupts complex function, and this can sometimes be corrected by coexpression of other members of the complex (CLARK-ADAMS et al. 1988). Indeed, we found that tub>sloth2, sloth1 animals were viable (Figure 8G) and exhibited normal bristles (not shown). Similarly, overexpression of the entire bicistronic transcript had no obvious phenotypes (Figure 8G, not shown).

In all cases in this paragraph, “(not shown)” refers to the wild-type or short scutellar bristle phenotype. The authors consider the reporting of bristle length to be useful qualitative information and the “(not shown)” was simply deleted. Examples of normal and short bristle length are reported in Figure 2.

“In addition, neuronal expression of sloth1 or sloth2 was unable to rescue mutant lethality (not shown).”

This “(not shown)” was an oversight and the information is reported in a revised Figure 2.